# Misspecified models create the appearance of adaptive control during value-based choice
Harrison Ritz [1,2,3] ✉, Romy Frömer [1,2,4,5] & Amitai Shenhav [6,7]

Decision scientists have grown increasingly interested in how people adaptively control their decision making, exploring how metacognitive factors influence how people accumulate evidence and commit to a choice. A recent study proposed a novel form of such adaptive control, whereby the values of one's options contribute to both the formation of a decision and the effortful invigoration of a response. In this framework, the control process was operationalized in a drift diffusion model as the lowering of the decision threshold on difficult trials. Reanalyzing the data from this experiment, we establish alternative explanations for these findings. We show that the reported evidence for controlled threshold adjustments can be explained away by task confounds, time-dependent collapses in decision thresholds, and stimulus-driven dynamics in an alternative form of evidence accumulation. Our findings challenge the specific evidence for this new theory of motivated control while at the same time revealing paths and pitfalls in computational approaches to a more general understanding when and how control guides decision-making.

Recent years have seen a growing interest in the intersections between value-based decision-making and cognitive control[1], providing researchers with fresh new insights into how people decide how to decide. These lines of research have demonstrated that decision-making is not a purely stimulus-driven process—one in which evidence accumulates based entirely on the values of one's options—but, rather, that evidence accumulation is subject to adaptive and online control[1]. A recent experiment[2] leveraged these new accounts of cognitive control over decision making, and their hypothesized neural correlates, to compare imputed neural predictions from theories of control versus learning. Building on behavioral evidence for a novel cognitive control mechanism, these authors argued that neural responses in the dorsal anterior cingulate cortex (dACC) are better captured by learning models than by their proposed control model. Here, we question the evidence provided for this novel control mechanism, and the validity of the neural predictions that were predicated on this evidence.

There is good reason to think that people exert cognitive control over their decisions. For example, a substantial body of work has demonstrated that people can adjust their decision threshold (how certain they must be before committing to a choice) by leveraging meta-cognitive information about the decision process[3–7]. Decision-makers increase this threshold when they have conflicting evidence in support of multiple options (to buy additional time to resolve the conflict and avoid making a mistake[3]), and they lower (collapse) this threshold over the course of a decision when there is an impending deadline (to ensure that some response is made in time[4–7]). One theory of control allocation, the Expected Value of Control (EVC) theory, proposes that such threshold adjustments can be understood as arising from a cost-benefit optimization process involving the dACC[8,9].

Vassena and colleagues[2] sought to interrogate the EVC theory as an account of dACC function by developing and testing a new variant of this model. To do so, they extended EVC to account for a previously untested form of threshold control, to which the EVC theory had yet to be applied. The authors proposed that when people make choices under a time deadline, they use the values of their options to make two kinds of decisions. One of these decisions involves accumulating evidence for which option is better, until they reach a decision threshold (i.e., classical evidence accumulation). The second decision involves determining whether the values of their options merit 'invigorating' a specific response. These predictions were formalized in a computational model of

[1]Cognitive, Linguistic, and Psychological Sciences, Brown University, Providence, RI, USA. [2]Carney Institute for Brain Sciences, Brown University, Providence, RI, USA. [3]Princeton Neuroscience Institute, Princeton University, Princeton, NJ, USA. [4]School of Psychology, University of Birmingham, Birmingham, UK. [5]Centre for Human Brain Health, University of Birmingham, Birmingham, UK. [6]Department of Psychology, University of California, Berkeley, CA, USA. [7]Helen Wills Neuroscience Institute, University of California, Berkeley, CA, USA. ✉e-mail: hr0283@princeton.edu

choice:

$$\log \frac{P(\text{response} = 1|\text{effort})}{P(\text{response} = 2|\text{effort})} = \alpha(V_1 - V_2) + \beta(\text{effort})$$

$$\text{effort} = \underset{\theta}{\arg\max} \; P(\text{response} = 1|\theta)V_1 + P(\text{response} = 2|\theta)V_2 - \|\theta\|$$

(1)

Where 'V' indicates the response value, and response-specific 'effort' depends on both the expected value of that response, and the required effort costs[10]. In this new EVC model, the cost-benefit analyses concerning how to intervene in a decision process (i.e., allocate effort to invigorate a response) depend on having an estimate of which response has higher value.

To operationalize these 'decision' and 'control' pathways, the authors examined how choice difficulty influences two components of a standard decision-making model. They modeled the decision pathway through participants' rate of evidence accumulation ("differences in integration of information without the need for control"[2]) and the control pathways through participants' decision threshold ("the need for subjects to decrease the decision threshold (that is, invigorate a response)"[2]).

The authors predicted that participants would use cognitive control over their threshold to ensure that the decision was made before the response deadline. When choices were easy (large difference in option values), the authors predicted that decision-makers should maintain a higher threshold. When choices were hard, however, they predicted that a person would lower their decision threshold and make a more immediate decision, ensuring a response before the deadline. This threshold-control model was central to the authors' account of EVC, justifying their imputed EVC predictions for model-based neuroimaging analyses (though see ref. 11).

While the value-driven threshold control described by these authors shares properties with forms of threshold control that EVC has been previously applied to (e.g., in response to expected deadlines, levels of decision conflict, or surprise), their proposed control mechanism differs from previous EVC work in two critical respects. First, it assumes that people adjust decision thresholds based on the specific values of their options, rather than based on the environmental statistics (i.e., agnostic to the properties of each specific choice) or metacognitive assessments (e.g., their uncertainty in the choice process as a whole, rather than specific option values). As we return to later, this assumption faces normative challenges that have not been previously addressed (i.e., why control the decision process if you already know the right answer?). Second, this model makes the *opposite* prediction from existing conflict-based metacognitive accounts, predicting that difficult choices should entail *decreased* thresholds rather than the increases in threshold previously observed in such conditions (compare to refs. 3,12, 13).

To test this novel account, the authors had participants perform a value-based decision-making task while undergoing fMRI. On each trial, participants selected between two options under a tight response deadline. Analyzing choice behavior from this task using a drift diffusion model (DDM[14,15]), the authors confirmed their prediction that participants set lower decision thresholds for difficult choices compared to easy choices. After validating the behavioral predictions of their EVC model, the authors then extracted neural predictions from the model (though such neural translations are also controversial[11]). When they compared these putative EVC predictions to neural activity observed in dACC, they found that dACC responses were more consistent with a learning-based account of dACC previously developed by the authors (the PRO model[16]) than with EVC.

Here, we revisit the behavioral evidence that was used to validate this novel variant of the EVC model.

## Methods
### Task design
Vassena and colleagues (2020) had participants perform a deadlined value-based decision-making task for monetary reward (Fig. 1A). On each trial, participants had a limited time window (750 ms) to choose between two

bundles, or they would miss out on their choice for that trial. Each bundle contained a pair of fractals, and participants had learned the fractal values during training. Each fractal was worth between €0.10 and €0.80 (in increments of €0.10), and participants received the value of one fractal from the pair they chose, chosen randomly, in addition to their participant payment. For instance, the left bundle could have fractals worth 70 and 90 points, and the right bundle could have fractals worth 30 and 40 points. After they chose their preferred bundle, participants received a monetary bonus based on the point value of a random fractal from the bundle. Participants were therefore incentivized to choose the more valuable pair within the time allotted, and it was better for them to make a choice than miss the deadline (i.e., options were entirely in the domain of gains). Participants performed 160 trials, which were balanced with respect to the total value and difference between the left/right options. Additional details on this task are available in the original manuscript (Vassena et al.[2]).

### Participants
Twenty-three participants (13 females; $M_{age} = 23$) were recruited, in accordance with the local research ethics committee. Twenty-two participants actually took part in the experiment. For exclusion criteria, and additional detail on demographics, please refer to the original manuscript[2].

### Drift diffusion analysis
We fit drift diffusion models (DDMs) that both matched the authors original claims, and models that explored alternative explanations for their results. All models were fit using HDDM[15,17], using five chains of 6000 samples (burning the first 2000 samples from each repetition). Model convergence was verified using the Gelman-Rubin R-hat statistic[18].

Apart from the Original model, models used the Likelihood approximation network (LAN) option in HDDM, providing efficient approximate likelihoods for collapsing bound DDMs. The Original model used the 'HDDM' function to predict correct responses, and the rest of the models used the 'HDDMnnRegressor' function to predict left/right choices. All models were fit with parameters at both the subject and group levels. As a requirement of the LAN fitting procedure, we used uninformed priors.

Models were compared using divergence information criteria (DIC[19]). Note that the total number of trials differs between the Original and Original* models, as the Original model removed the most difficult choices. Normalizing the deviance by the sample size ('trial-averaged likelihood') still resulted in a DIC difference of 11.1 in favor of the Original* model, with the Original* model also offering a more comprehensive account of participants' behavior. Parameter posteriors and posterior predictive checks were plotted using (Gaussian) kernel density estimation in MATLAB.

We validated our model-fitting and model-selection processes using model recovery (Supplementary Fig. 3). We generated forty synthetic datasets from three models (Original*, VD static bound, VD collapsing bound) by sampling from the estimated parameter distributions for each participant. We then fit these three models to each generated dataset. To better match across fitting procedures, we fit all three models using HDDM's likelihood approximation networks (though we found qualitatively similar results using the same fitting procedure as in the main experiment). We found that our model comparison procedure was biased towards collapsing bound models, with better DIC even when the data were generated by a static bound model. However, these biases were an order of magnitude smaller than the empirical differences in DIC that we found when we fit these models to participants. Given how unlikely these empirical differences were under the null model generated by model recovery, model recovery supports the conclusions in the main text.

We also validated that our estimated parameters were identifiable using parameter recovery. We generated forty synthetic datasets by sampling from the estimated parameter distributions for each participant (Supplementary Fig. 5). Fitting the best-fitting model to this synthetic dataset, we found that there was a good correspondence between the generating and estimated

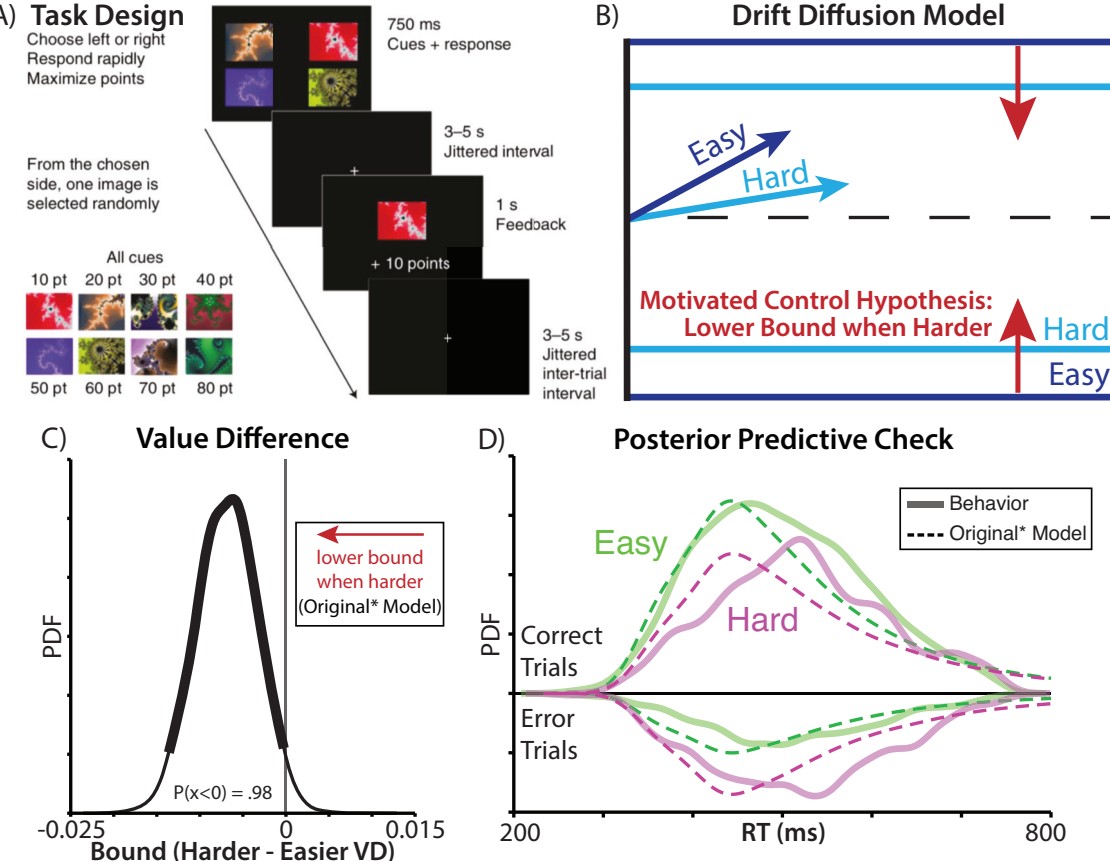

**Fig. 1 | Predictions of the motivated control account are confirmed when specifying the drift diffusion model as in the original study. A** In this experiment, participants chose between two bundles of fractals worth a known amount of money. Time pressure was introduced by requiring that choices be made within a 750 ms deadline, otherwise forfeiting any reward for that trial. **B** The motivated control account predicts that harder choices (those involving similarly valued bundles; dark blue) should be associated with lower rates of evidence accumulation (as predicted by other accounts) and, critically, that these choices should also be associated with lower decision thresholds, relative to easy choices (light blue). **C** Consistent with this account, when fitting drift diffusion models to these data using similar specifications as in the original paper, we find that harder choices are reliably associated with lower thresholds (97.5th percentile of posterior density was less than 0). **D** When comparing RT distributions predicted by this model (dashed lines) to the actual data (solid lines), we find that the predicted distributions are substantially more skewed than the actual ones, both for easy (green) and hard (purple) choices. See also Supplementary Fig. 1. Panel A adapted from (Vassena et al.[2]).

---

parameters, supporting their identifiability, and our ability to interpret the parameters. Table 1 summarized all models that were fitted and compared.

**Leaky competing accumulator (LCA)**

The LCA is a biologically-plausible decision model, in which evidence in each accumulator (y) evolves according to the input drive (I; from option value V), accumulator 'leak' (k), mutual inhibition (w), and IID noise ($\sigma$), before being passed through a rectified linear unit (ReLU):

$$y_{t+1}^1 = ReLU\left(y_t^1 + dt\left(I^1 - ky_t^1 - wy_t^2\right) + s_t^1\sqrt{(dt)}\right) \quad (2)$$

$$y_{t+1}^2 = ReLU\left(y_t^2 + dt\left(I^2 - ky_t^2 - wy_t^1\right) + s_t^2\sqrt{(dt)}\right) \quad (3)$$

$$I = b\left(mV^{\max} + (1-m)V^{\min}\right) \quad (4)$$

$$s_t \sim N(0, \sigma) \quad (5)$$

$$y_0 = 0 \quad (6)$$

This LCA model had 8 parameters: non-decision time (positive), total input drive (positive), ratio between max-value and min-value input drive (0-1), accumulator leak (positive), mutual inhibition (positive), initial

bound (positive), boundary collapse rate (positive), and noise (positive). The timestep (dt) was fixed to 0.005 for fitting and 0.001 for sampling. For simplicity, we accuracy-coded the data such that y1 corresponded to the correct choice, and y2 corresponded to the incorrect choice.

Since there isn't a closed-form solution to the LCA likelihood, we fit our model to participants' group-level behavior using simulation-based inference. First, we removed the between-subject variance from participants' RTs (subtracting each participant's mean and then re-adding the grand mean), to prevent RT quantiles from being dominated by individual differences. We then computed the RT deciles for correct and incorrect trials on trials with high vs low overall value (median split), with separate bins for deadline-missing trials. We conditioned the RT deciles on overall value to provide more information about the RT distribution, while using a contrast that was orthogonal to the objective value difference (the original hypothesized factor).

We computed the data likelihood on each trial using inverse binomial sampling (IBS[20]). IBS provides an unbiased and sample-efficient estimate of stochastic log-likelihoods based on counting the number of samples to reach a match between a simulated and observed trial category (here, the RT bin). We maximized this log-likelihood with Bayesian adaptive direct search (BADS[21]), a derivative-free optimization procedure that leverages IBS's estimates of the uncertainty in the log-likelihood.

To compare the model fits between LCA and DDMs, we computed a quantile-based Bayesian Information Criterion (BIC; Ratcliff & Smith[22])

**Table 1 | Model overview**

| Model Name | Model Type | Regression Equation | Additional parameters |
|---|---|---|---|
| Original | ddm | v ~ [hard, easy]<br>a ~ [hard, easy]<br>z ~ [hard, easy] | z |
| Original* | ddm | v ~ -1 + signVD,<br>a ~ 1 + absVD | z, p_outlier |
| VD | ddm | v ~ -1 + maxVD + minVD,<br>a ~ 1 + absMaxVD + absMinVD | z, p_outlier |
| OV | ddm | v ~ -1 + maxVD + minVD,<br>a ~ 1 + maxOV + minOV | z, p_outlier |
| VDOV | ddm | v ~ -1 + maxVD + minVD,<br>a ~ 1 + absMaxVD + absMinVD + maxOV + minOV | z, p_outlier |
| VD both | angle | v ~ -1 + maxVD + minVD,<br>a ~ 1 + absMaxVD + absMinVD,<br>theta ~ 1 + absMaxVD + absMinVD | z, p_outlier, theta |
| VD init | angle | v ~ -1 + maxVD + minVD,<br>a ~ 1 + absMaxVD + absMinVD | z, p_outlier, theta |
| VD rate | angle | v ~ -1 + maxVD + minVD,<br>theta ~ 1 + absMaxVD + absMinVD | z, p_outlier, theta |
| OV both | angle | v ~ -1 + maxVD + minVD,<br>a ~ 1 + maxOV + minOV,<br>theta ~ 1 + maxOV + minOV | z, p_outlier, theta |
| OV init | angle | v ~ -1 + maxVD + minVD,<br>a ~ 1 + maxOV + minOV | z, p_outlier, theta |
| OV rate | angle | v ~ -1 + maxVD + minVD,<br>theta ~ 1 + maxOV + minOV | z, p_outlier, theta |
| VDOV both | angle | v ~ -1 + maxVD + minVD,<br>a ~ 1 + absMaxVD + absMinVD + maxOV + minOV,<br>theta ~ 1 + absMaxVD + absMinVD + maxOV + minOV | z, p_outlier, theta |
| VDOV init | angle | v ~ -1 + maxVD + minVD,<br>a ~ 1 + absMaxVD + absMinVD + maxOV + minOV | z, p_outlier, theta |
| VDOV rate | angle | v ~ -1 + maxVD + minVD,<br>theta ~ 1 + absMaxVD + absMinVD + maxOV + minOV | z, p_outlier, theta |

*Model types:* ddm = static bound, angle = linear collapsing bound. *Regression Equation:* v = drift rate, a = initial threshold, theta = collapse rate. *signVD* = signed value difference (equal weight), *absVD* = absolute value difference (equal weight). max/min VD = value difference for the maximum or minimum value options (e.g., max reflects the highest value left option and the highest value right option). max/min OV = summed value for the max or min value options. *Additional parameters:* z = starting point, p_outlier = lapse rate, theta = collapse rate.

based on 100 datasets simulated from each fitted model. For LCA, we used the same simulator that was used for fitting. For DDMs, we refit our Original* and (best-performing) 'VDOV-both' DDMs to the same group-level dataset that we used for the LCA (i.e., with between-subject variance removed). Note that the DDM simulations were sampled from the posterior parameter distribution, which likely biases towards worse BICs relative to just sampling from the best point estimates (as was done with LCA). To militate against this concern, as a more conservative test, we also confirmed that the best LCA simulation still outperformed the best DDM simulation ($\Delta$BIC = 104; 8% of LCA simulations had better BICs than the best VDOV-both simulation). We note also that the goal of these simulations is to provide evidence that a control-free LCA—which has been shown to be a more biologically plausible model than the DDM, and better able to account for several characteristics of decision-making —provides a sufficient and parsimonious account of choice behavior from this task.

To compute the BIC, we evaluated the frequencies of simulated trials in each decile of the empirical RT distribution, for correct and error trials in easy and hard value-difference conditions (which the DDMs were explicitly conditioned on, but not LCA). The BIC for each simulated dataset was computed as:

$$BIC = -2\left(\sum_{i}^{10} Np_i \ln(\pi_i)\right) + M \ln(N) \qquad (7)$$

Where N is the sample size, $p_i$ is the probability of the empirical data in each decile, $\pi_i$ is the probability of the simulated data in each decile, and M is the number of free parameters (LCA: 8, Original*: 5; VDOV-both: 14). BIC explicitly penalizes model complexity by counting the number of free parameters (second term). BIC-differences greater than 2 are considered meaningful and differences greater than 6 and 10 are considered strong and very strong evidence for the best model, respectively[23].

To fit DDMs to the LCA model, we concatenated three LCA-simulated datasets (reaching ~10,000 trials), and then fit the Original* and VDOV-both DDMs at the group level using HDDM.

Data distribution was assumed to be Gaussian for parametric statistical tests; however, this was not formally tested.

Neither the original study, nor the present reanalysis of the data was formally pre-registered; however, several of the core concerns about the original study that we report here were proposed in an earlier publication[12].

## Reporting summary
Further information on research design is available in the Nature Portfolio Reporting Summary linked to this article.

## Results
### Models with motivated control (value-based threshold) leave room for improvement
The original authors tested their prediction that people would invigorate responses on difficult trials by fitting a hierarchical drift diffusion model (HDDM) to participants' accuracy and reaction times. They fit separate drift

rate, threshold, and starting point parameters for easy and difficult trials, defined based on the absolute difference in bundle value (weighting the fractals equally; Fig. 1B). When choices were more difficult, they found that drift rates were lower, consistent with weaker evidence for one option relative to another[3]. Critically, consistent with their predictions, they *also* found that participants' thresholds were lower during more difficult trials (percent of posterior density below zero: $P(x < 0) = 0.98$).

Using the same HDDM model specifications provided by the original authors ('Original Model'), we replicated all the findings reported above, most notably that drift rate and threshold vary with value difference in the predicted direction (Supplementary Fig. 1A). However, these analyses also suggested that there was significant room for improvement in the model's ability to capture choice behavior on this task: model-generated choices and response times, as originally specified, failed to provide a qualitative match to the empirically observed choice and RT distributions (Supplementary Fig. 1B). For instance, their best-fit model predicted RT distributions that were substantially more skewed than the actual data.

As initial steps towards addressing these concerns of model fit, we generated a revised model that encompassed the spirit of the author's original predictions, but included modest adjustments to better account for participants' choice behavior ('Original* Model'). First, to account for potential response biases (e.g., favoring the left or right option), we coded choices in terms of which side was chosen (i.e., 'response-coded' rather than 'accuracy-coded') and fit subject-specific starting points. The starting point now determined the left/right biases, and not a bias towards the correct/incorrect choice (before any evidence had accumulated) like in their original model. Since choices were response-coded, we included the most difficult trials, which were omitted in the original analysis. Second, we included an outlier term to account for attentional lapses. Third, we fit difficulty as a continuous variable, rather than a discrete variable, as the psychological construct is usually thought of as continuous and median splits (as used in the Original model) can reduce statistical power.

These model adjustments collectively led to quantitative improvements in model fit (change in DIC between Original and Original* model = 569), without altering any of the key observations: with this elaborated model, we still found that thresholds were significantly lower for difficult relative to easy choices (Fig. 1C). However, when simulating distributions of choices and response times from this model, we found that these distributions still provided a poor qualitative match to empirically observed behavior (Fig. 1D), predicting a much more heavily skewed RT distribution. This mismatch between model and data motivated us to consider additional revisions that would better account for core elements of the task and findings.

**Evidence for motivated control is eliminated when accounting for core elements of choice**

Based on the design of the choice task and the patterns of behavior that were reported by the original authors, we considered three modifications to the elaborated model above, and tested each one in turn.

First, the models above assume that participants weighted all four fractals equally when choosing between the two bundles. As a consequence, value difference (and thus choice difficulty) is determined by comparing the average value of each bundle. However, this assumption is contradicted by the authors' regression analyses showing that participants over-weight the higher-value fractal within each bundle relative to the lower-value one (placing five times as much weight in the former than the latter)[2]. We therefore tested whether separately modeling the relative and overall influence of the higher-valued and lower-valued options in each pair improved model fit.

Second, the models above only consider how the parameters of the DDM will be influenced by the *difference* between the two option values. However, follow-up analyses by Vassena and colleagues[24] report that these parameters also differ by *overall* set value (sum of left and right option values), consistent with a broader array of value-based decision-making

research[25–28]. While the authors had originally designed their bundles to orthogonalize overall value and value difference, this was done under the assumption that participants weighed the fractals in each bundle equally. We reevaluated this assumption based on the unequal weights revealed in the analyses above, finding that these re-weighted estimates of overall value and value difference were now correlated (Supplementary Fig. 2). We therefore tested whether DDM fits would be improved by including overall value as a predictor of drift and threshold in our model (along with value difference).

Third, the models above assume that participants maintained a fixed response threshold throughout the choice period. Past work suggests that decisions under time pressure are instead often better characterized by a response threshold that decreases (collapses) over time, requiring less evidence over time but guaranteeing that a response is made before one's deadline[5] (Fig. 2A). A collapsing threshold model differs from the motivated control account above in that it is agnostic to the value of one's options on a given trial, instead assuming that thresholds collapse at the same rate on every trial (e.g., due to the deadline[4,7,29]). In other words, a collapsing threshold, in and of itself, does not necessitate trial-to-trial adjustments to motivated control. We tested whether a collapsing threshold improved model fit relative to a fixed threshold model.

We found that models incorporating these modifications fit participants' data far better than the Original* model (Fig. 2B). A model allowing weights to differ between max and min values fit better than the original equally weighted model ($\Delta$DIC = 119). Models with collapsing bounds substantially outperformed models with static bounds (Value difference [VD] model: $\Delta$DIC = 1206, Overall value [OV] model: $\Delta$DIC = 1243, model with both [VDOV]: $\Delta$DIC = 1184). Critically, unlike models with static bounds, models with collapsing bounds were able to accurately capture the skew of participants' reaction time distributions (compare dotted and dashed lines in Fig. 2C). Both in models with static bounds and collapsing bounds, models that included an influence of overall value on threshold fit better than models in which overall value was excluded (static-bound VDOV vs. VD: $\Delta$DIC = 107; collapsing-bound VDOV vs. VD: $\Delta$DIC = 85).

We validated our model selection procedure using model recovery. We generated synthetic datasets from three models (Original*, VD static bound, VD collapsing bound) by sampling from participants posterior parameter distributions, and then we fit each model to each generated dataset. We found that our model comparison procedure was biased towards collapsing bound models, with better DIC even when the data were generated by a static bound model (Supplementary Fig. 3A). However, these biases were an order of magnitude smaller than the empirical DIC differences from fitting these models to participants (Supplementary Fig. 3B). Given how unlikely our empirical DIC differences were under the null distribution generated by our model recovery, as well as the striking differences in posterior predictive checks (Fig. 2C), these findings continue to support the collapsing bound mechanism. Speculatively, our use of participants' parameters may have found a region of the parameter space where there was particularly high mimicry between static and collapsing bound models. Another alternative that has been proposed to the collapsing bound mechanism is across-trial variability in drift rate[30]. However, we found that models with across-trial variability, but not collapsing bounds, provided a poor account of the RT distribution (Supplementary Fig. 4A).

We validated the identifiability of these parameter estimates using parameter recovery in our best-fitting model: simulating datasets with ground-truth parameters, re-fitting the model to these datasets, and then measuring the similarity between ground-truth parameters and estimated parameters. We found that we had strong parameter recovery (Supplementary Fig. 5), improving our confidence in these parameter estimates.

Importantly, to the extent difficulty was associated with threshold adjustment, it was now in the exact opposite direction from what Vassena et al. described. Harder choices were now associated with *higher* rather than lower thresholds (Fig. 2D), contradicting the predictions of their motivated control account. Instead of difficulty, the strongest influence on threshold was the overall value of the choice set (Fig. 2E; see Supplementary Fig. 5 for

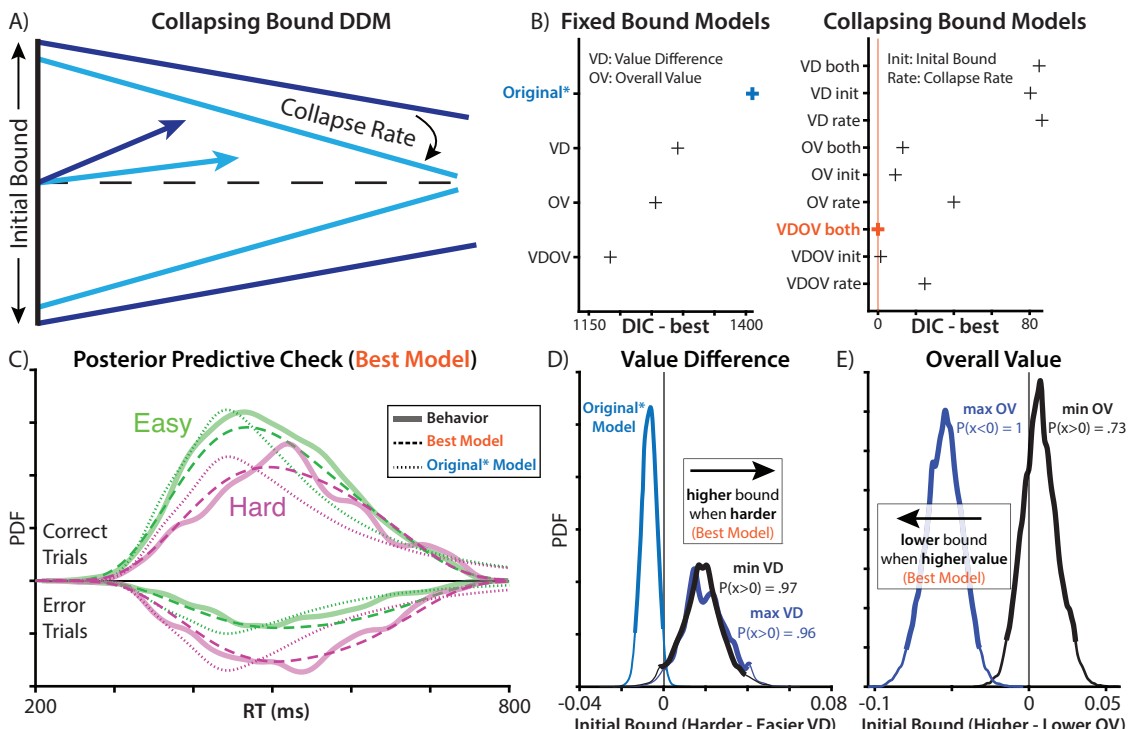

**Fig. 2 | Elaborated models improve fit and contradict original findings. A** To account for the response deadline, we fit a DDM with a threshold that linearly decreases (collapses) over time rather than remaining fixed over an entire choice (as in the previous models). **B** Complexity-penalized model comparisons show that incorporating this collapsing bound produces substantial improvement in model fit. Model fit further improves when allowing this collapsing bound to be further modulated by both value difference and overall (summed) value of the options, as well as when allowing for separate weights on the higher- and lower-valued fractals within each pair. **C** Simulated behavior from the best-fitting model (dashed line) provides a substantially better qualitative fit to participant data (solid line) than the Original* model (dotted line; cf. Fig. 1D). **D** Under this best-fit model, we no longer see difficulty associated with decreases in threshold, but instead a trend in the reverse direction (compare Fig. 1C). **E** This model also shows that higher overall value is associated with lower thresholds, consistent with a separate prediction of the motivated control account.

---

all group-level parameters), which is more consistent with previous work on value-based decision making[25–28].

### Evidence of motivated control can emerge as an artifact of model mimicry

Our findings so far undercut one major prediction of the motivated control account, which is that people adjust their threshold downward on difficult trials (i.e., to invigorate responses based on option values). However, these models retain clear evidence for a revised account[2,11] in which people use information about the overall option value to lower their threshold (Fig. 2E). This leaves open the possibility that participants are engaging in a form of motivated threshold-adjustment, but one that is more sensitive to overall value than choice difficulty. However, previous research points to an explanation of these findings without invoking any form of motivated control.

Prior work has found that value-dependent speeding can naturally emerge from a stimulus-driven evidence accumulation process. In the leaky competing accumulator (LCA[31]; Fig. 3A), more valuable options naturally produce faster decisions for higher-value choice sets (see also[25,26,28,32]). Unlike in the DDM, which accumulates the difference in option values (regardless of the overall value of those options), LCA response units are independently driven by inputs and compete through mutual inhibition (see also refs. 33,34). As a consequence, the total activation across accumulators rises faster for higher value option sets, and activation-dependent mutual inhibition causes a faster divergence between accumulators, both resulting in faster response times. Critically, this dynamic occurs *without* any additional control processes for adjusting decision thresholds based on option values. It is therefore possible that the apparent value-related control of threshold in the DDM could be mimicking a control-free decision architecture.

To test whether the LCA could plausibly account for participants' behavior, we fit an LCA model to participants' behavior. As in high-performing DDMs, we fit separate weights for high- and low-value options, as well as a linearly collapsing bound. Unlike the previous DDM-based models, however, value was not otherwise able to influence the decision threshold, with the initial bound and collapse rate instead held constant across all trials.

Since the LCA does not have an analytic likelihood function, we fit this model to participants' group-level reaction-time distributions using simulation-based inference (see Methods). Posterior predictive checks revealed that the fitted LCA model was indeed a good match to participants' behavior (Fig. 3B). To quantitatively assess the quality of this fit, we compared a complexity-penalized fit metric (quantile-based Bayesian Information Criteria BIC[22]; see Method) between the LCA and two data-matched DDMs (Original* and the best-fitting DDM; refer to Fig. 2B). Comparing BIC across 100 simulations of each model, we found that the LCA provided an improvement over the best-fitting DDM (difference in mean BIC: bootstrap 95% CI = [185, 277]; Fig. 3C).

The fact that this "control-free" model could closely reproduce behavior indicated that stimulus-driven dynamics were sufficient to account for choice behavior, without needing to invoke motivated control. It also suggested that the value-related threshold adjustments in the DDM analyses may have been illusory. To explicitly test this hypothesis, we simulated choice behavior from our LCA model and fit it with our DDMs (Fig. 2A). Remarkably, we found that the LCA threshold was estimated to significantly decrease when overall value was higher, despite the *absence* of such a connection in the generative model (Fig. 3D). Thus, overall-value-dependent threshold adjustments (within a DDM framework) artifactually mimicked the true effect of overall

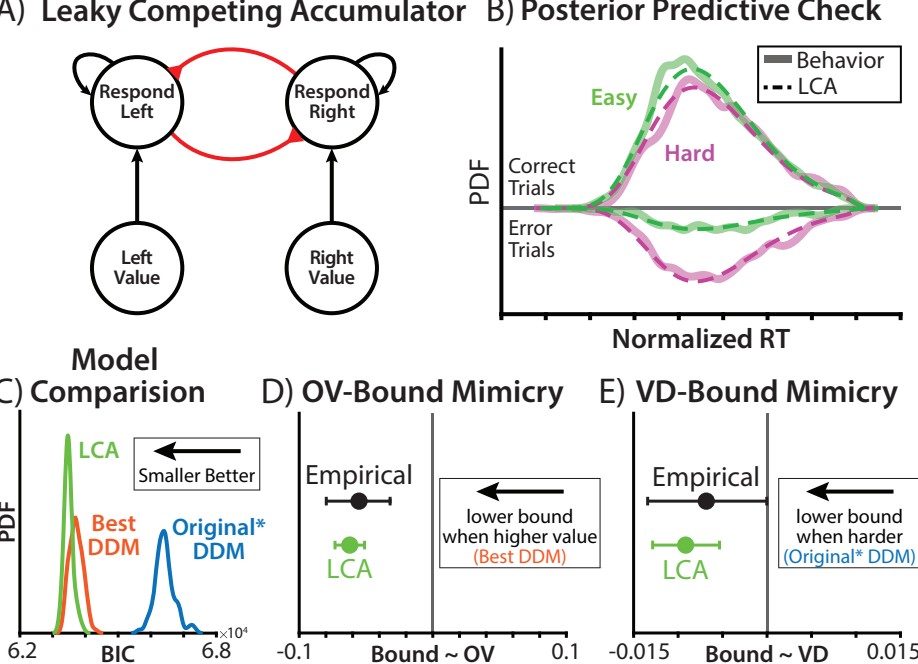

**Fig. 3 | A control-free model can account for choice behavior and mimic motivated control.** **A** To test whether stimulus-driven processes were sufficient to account for apparent evidence of motivated control, we fit a leaky competing accumulator (LCA) that lacked a value-based mechanism for threshold control. Black connections indicate excitation, red connections indicate inhibition. **B** The fitted LCA provided a good account of participants' behavior (refer to Fig. 2C for plotting convention). Because the LCA was fit at the group level, 'Normalized RT' removes between-participant variance (as in the fitted data), and is normalized by the response deadline (for visualization). **C** Bayesian information criteria (BIC; complexity-penalized fit metric) for three group-level models. The distribution of BICs reflects the variability across the model simulations used to compute the BIC. **D, E** DDMs were fit to simulated behavior from the fitted LCA and compared to the participant-estimated coefficients. Error bars indicate 95% credible intervals. **D** Posterior estimates of the relationship between initial bound and the overall value (OV) of the max-value pair. 'Empirical' indicates the DDM fit to participant data (refer to Fig. 2E). **E** Posterior estimates of the relationship between the decision bound and the value difference (VD) between options. 'Empirical' indicates the DDM fit to participant data (refer to Fig. 1C).

value driving choice through accumulator dynamics (within an LCA framework).

The other evidence for motivated control, from the original experiment, was lower thresholds for more difficult choices. Greater difficulty was also associated with weaker evidence accumulation, in all of the models we've described. This means that faster vs slower decisions would fit the threshold at higher vs. lower points in its collapse, respectively. Thus, even though that collapsing threshold is only driven by time within a trial (and not option value), the threshold level at which a decision is reached differs based on difficulty. If a model omits this collapsing mechanism and instead assumes that thresholds are fixed within a trial as was the case for Vassena et al.'s models[2,24]—then it may appear as though trials with lower value differences also have lower fixed thresholds. Moreover, confounding between overall value and value difference under unequal weighting (see Supplementary Fig. 2) only exacerbates the potential for an artifactual influence of value difference.

To test whether difficulty-dependent control could artificially occur in a stimulus-driven process with collapsing bounds, we fit the authors' DDM (Original*; with static bounds) to LCA-generated behavior. We found that this DDM mistakenly identified reduced thresholds for harder choices (Fig. 3E), despite there being no such mechanism in the generative process. Thus, a model that lacks motivated control over decision thresholds can not only account for behavior on this task, but can easily generate artifactual signatures of control.

## Discussion

Decision scientists have grown increasingly interested in how we control our decision-making[1,35,36]. Building off of research into adaptive control mechanisms in other forms of cognitive tasks[37–39], studies have shown that such decision parameters can be shaped by variables such as temporal urgency[5] and choice conflict[3,12,13]. Vassena and colleagues contributed a new model of threshold control to this developing field, extending the algorithmic framework offered by the EVC theory. This new variant of the EVC model uses information available during a choice (option values) to guide adaptations to that choice (threshold adjustments). The authors provided behavioral evidence for this account by demonstrating that difficult speeded

decisions were associated with lower decision thresholds than easy ones, consistent with adaptive reductions in threshold to avoid the possibility of failing to choose either option.

We reexamined their findings with a more comprehensive modeling approach, and showed that this apparent value difference-related threshold decrease arose artifactually from misspecifications in the original choice model. When all features of these speeded decisions were properly accounted for, hard decisions were if anything associated with higher rather than lower thresholds. We further showed that these value-driven threshold adjustments could be accounted for by a control-free decision process (LCA).

### Control mimicry in value-based decision making

Our findings provide alternative explanations for putative control signatures that were used to justify the authors' model-based fMRI analyses. Moreover, these findings highlight important opportunities and pitfalls for future research into related mechanisms of adaptive control. We begin by addressing the latter before returning to the former. We show that even for relatively simple choices, between two bundles of learned values, there are at least three major modeling decisions that can substantively impact researchers' conclusions (Fig. 4).

First, we showed that model fits can be improved by allowing the values in each option pair to have separate weights in the decision rather than averaging these uniformly (Fig. 4A). These differential weights were evidenced in behavior, and are consistent with previous work suggesting differential attentional priority under constrained time and attentional resources[40]. Failing to account for these asymmetric weights created the false appearance that value difference and overall value were orthogonalized across choices (as originally intended), whereas accounting for these asymmetries reveals that these variables were in fact correlated. Thus, results previously attributed to value difference (e.g., its predicted influence on threshold) were at least in part contaminated by effects of overall value.

Second, we showed that model fits were improved by allowing thresholds to collapse over time within a trial (agnostic to information about the options themselves; Fig. 4B). This is consistent with a wide array of previous findings that implicate such dynamic time-based threshold

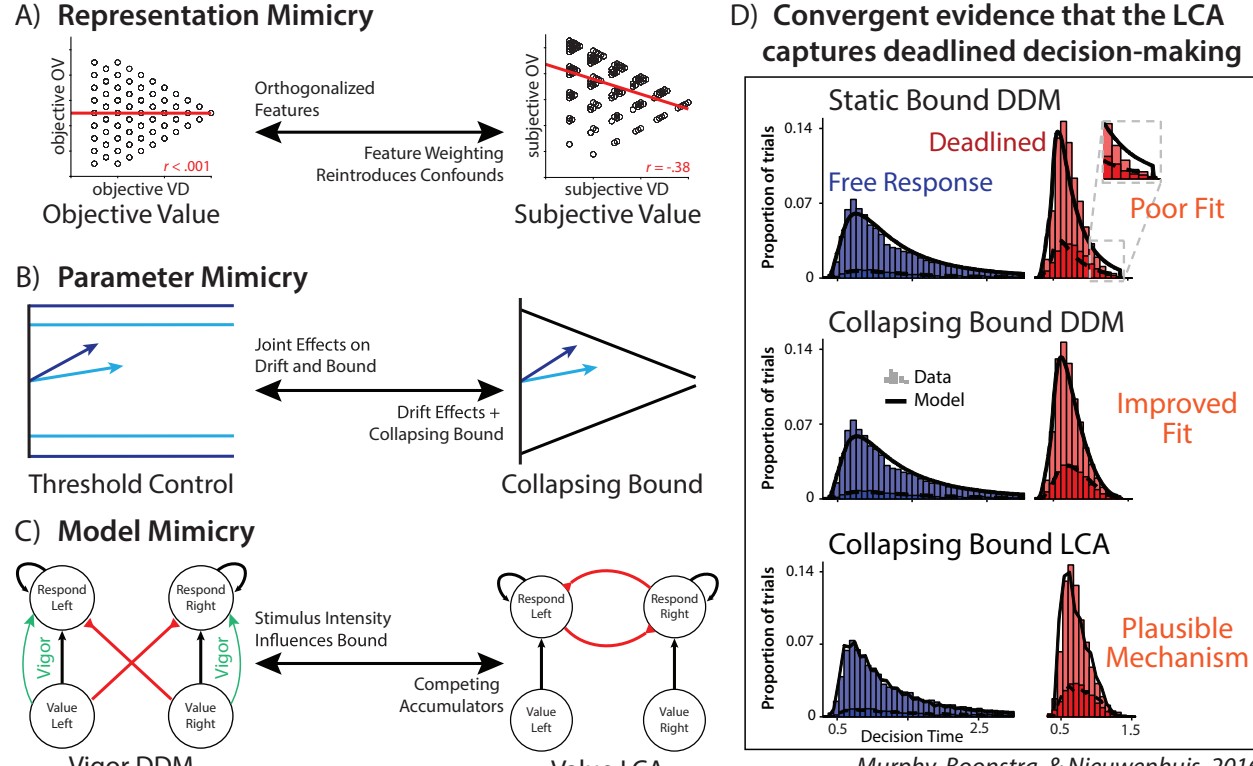

**Fig. 4 | Multiple sources of mimicry in modeling value-based choice.**
**A** *Representation mimicry* can occur if participants are using different information than the model assumes they use. In this experiment, participants' strategy over-weighting of high-value fractals relative to low-value fractals hid the influence of overall value on decision-making when this weighting strategy was unaccounted for.
**B** *Parameter mimicry* can occur when different parameterizations can capture similar behaviors. In this experiment, the joint effects of drift rate and threshold mimicked the influence of a collapsing bound[11]. **C** *Model mimicry* can occur when different model classes can capture the same behavior. In this case, the influence of overall value on decision threshold could be parsimoniously accounted for by intrinsic dynamics of an independent accumulator model such as an LCA (see

also[54]). Previous research in decision making provides guidance for navigating these mimicries. Previous work[29] has found that deadline conditions can increase 'urgency', leading to lower confidence-at-choice for slower decisions. **D** Modeling of speed-accuracy trade-offs in deadlined decision-making tasks[43] has revealed that collapsing bound DDMs offer useful statistical models, and collapsing bound LCAs can offer mechanistic models with biological plausibility. Theoretical modeling of value-based decision making[32] has found that LCAs are sensitive to the overall value of the choice set (see also refs. 25,54), creating mimicry if overall value is confounded with difficulty. Panel D adapted from Murphy, Boonstra, & Nieuwenhuis (2016; CC BY 4.0)[43].

adjustments[4,5,7], particularly in the presence of explicit time pressure as in the current study[4,5,7,29,41]. Failing to account for these collapsing bounds not only results in poorer model fits, but it can also fundamentally alter one's conclusions about other parameter estimates. Specifically, we show that collapsing bounds can be mimicked by joint changes in drift rate and threshold. As a result, variables that like value differences in the present study induce increases in drift rate will also artifactually increase threshold. Covariance among these parameters can therefore also serve as one diagnostic for latent threshold dynamics.

Third, we show that putative measure of value-dependent threshold adjustment within a DDM framework can mimic stimulus-driven evidence accumulation from other evidence accumulation frameworks (i.e., without positing an additional top-down mechanism for threshold control; Fig. 4C). While our simulations demonstrate this using the LCA, this form of mimicry generalizes to any other generative choice model that naturally gives rise to overall-value-related speeding effects, including other parallel accumulator models[26,42] and attentionally weighted DDMs[28].

The implications of these modeling pitfalls are not restricted to value-based decision tasks, like in the experiment under consideration here. Rather, their relevance to value-based decision making is already evident from existing research programs in perceptual decision making. For instance, Thura and colleagues[29] found that response deadlines induce participants to respond at lower levels of confidence as they approach the

deadline (modeled as enhanced 'urgency' signals, which are equivalent to collapsing bounds). Paralleling the findings of the current experiment, Murphy and colleagues[43] found that DDMs with condition-independent collapsing bounds effectively captured performance during deadlined random dot motion tasks (see also ref. 4; Fig. 4D). They proposed an LCA (with time-dependent gain for threshold collapse) could serve as a biologically plausible model of deadlined evidence accumulation. Within the domain of LCA modeling, there is a long history of integrating theories across perceptual and value-based decision making. While originally proposed as a model of perceptual decision making[31], subsequent work by Bogacz and colleagues[32] outlined the predictions of LCA for value-based decision making, including showing how the overall value of the choice set can produce faster reaction time in the LCA variant used here (see also ref. 25). Leveraging these inductive biases for value-based decision research can help guide researchers' modeling of under-constrained latent variables like cognitive control[1,44].

### Questioning the normative case for response invigoration
Collectively, this re-analyses undermine the empirical foundation for Vassena and colleagues' EVC model. While it cannot be ruled out that future studies will lend support for this account, this empirical gap encourages further reconsideration of the *normative* foundations of this account. Most notably, the model's proposed mechanism requires that information about options values (e.g., value difference) is available and used to selectively

invigorate responses (see Eq. 1). From first principles, there are problems with this requirement.

First, the values of the options are precisely what the decision-maker seeks to identify as they make their decision. To the extent the decision-maker has access to these values directly, the control process becomes obsolete. This same concern extends to our alternative DDMs. For similar reasons, existing models that account for online control adjustments do so not by relying on values per se, but instead on metacognitive representations that can be read out from the decision process (e.g., uncertainty or conflict[3,13,45,46]) or from longer-run statistics of the environment (e.g., reward rate[7,47]).

Second, Vassena et al.'s model predicts that, all else being equal, people will choose more quickly when choices are difficult relative to when they are easy. Virtually all empirical research, including the author's own data, finds the exact opposite to be the case, both in the case of perceptual and value-based decision-making (e.g. [48,49]). Third, the model draws inspiration (and intuitive support) from applications of adaptive control to decision-making in other cognitive tasks (e.g., flanker, Stroop). However, in these tasks control can be applied to enhance the signal associated with predetermined objects, locations, or features, based on the task rules that define relevant and irrelevant information (e.g., attend to ink color). By contrast, the response to a value-based decision—and hence which sources of information are relevant—is inextricably tied to the information being collected online, thus making the target of control a moving target, at best.

While our current findings focus on the computational mechanisms underpinning choice behavior, they nevertheless carry significant implications for understanding neural mechanisms driving such decisions. The original authors argued that activity in dACC during their task was inconsistent with predictions that they derived from their EVC model (see also ref. 11 for additional concerns about how neural predictions were extrapolated from the EVC theory). Our current findings raise a fundamental concern with this conclusion: their proposed motivated control account is unsupported by the findings from their experiment. Their critical finding, that people selectively invigorate difficult trials, can be explained away by task confounds and control-free decision architectures. This renders moot any neural predictions built on this evidence ("the predictions derived from the EVC model in our simulations are valid only if it can be demonstrated that our speeded decision-making task requires control"[2]), potentially explaining why Vassena and colleagues were unable to find evidence for these neural predictions. It remains to be determined whether previous adaptive control accounts can quantitatively account for patterns of dACC activity that were observed in this experiment. However, at a qualitative level, their fMRI results can be accounted for by any theory that implicates the dACC in monitoring for potential conflict and surprise, including EVC[1,8,11,50,51].

## Limitations

There are several limitations to our conclusions. First, we re-analyzed a single study with relatively few participants and few trials. While our findings are consistent with previous reports (e.g., [43]), applying these modeling techniques to larger and more diverse datasets may provide further confidence in, or refinements to, our conclusions. Second, our LCA simulations should be taken as a proof of concept that this model class could plausibly account for the originally reported findings. LCA models are difficult to fit due to their lack of an analytic likelihood, and are difficult to interpret due to parameter degeneracy[52]. Future work could compare the performance of our fitting procedure to alternative methods like neural network distillation[17], in order to provide the most accurate inference for this plausible model of value-based decision making. Finally, our re-analyses refute the cognitive basis for the original authors' interpretation of their neuroimaging findings. However, the current study does not re-analyze these neuroimaging data themselves, and cannot speak to their statistical validity.

## Conclusion

Substantial progress has been made toward uncovering the neural and computational mechanisms at the intersection of value-based decision-making and cognitive control. This research flourishes when bold new mechanistic accounts are proposed and tested. Recent examples of this include work that normatively accounts for and empirically demonstrates dynamic adjustments to evidence accumulation based on uncertainty and expected information gain[45,46,53], and dynamic adjustments in threshold based on conflict[3,12,13]. By providing a careful reexamination of both the empirical and normative basis of this recent and influential addition to this literature, we hope that we have both raised the threshold for adopting such an account, while at the same time accelerating progress towards more fruitful accounts of when, whether, and how we control our decisions.

## Data availability

Data is available from the original authors {Vassena, Eliana et al. [2]}.

## Code availability

Analysis code is available at https://github.com/shenhavlab/appearance-of-adaptive-control.

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

## Acknowledgements

This research was supported by NIH grant 1R01MH124849-01, NIH grant 2P50MH106435, NIH Shared Instrument Grant S10 OD025181, and NSF grant 2520024, an AMS Springboard award to RF, and the CV Starr Fellowship for HR. The funders had no role in study design, data collection and analysis, decision to publish or preparation of the manuscript. The authors are grateful to Sebastian Musslick, Jon Cohen, Matt Botvinick, and Michael Frank for valuable discussions, and to Alexander Fengler for essential advice on using the HDDM toolbox for fitting collapsing bound models.

## Author contributions

H.R., R.F., and A.S. conceived the study, H.R. performed the modeling, and all authors wrote and edited the manuscript.

## Competing interests

The authors declare no competing interests.
