## [Transparent Peer Review file · Communications Psychology]

Misspecified models create the appearance of adaptive control during value-based choice

Corresponding Author: Dr Harrison Ritz

Version 0:

Decision Letter:

Dear Dr Ritz,

Thank you for your patience during the peer-review process. Your manuscript titled "Phantom controllers: Misspecified models create the false appearance of adaptive control during value-based choice" has now been seen by 2 reviewers, whose comments are appended below. I have discussed the reports with my colleagues and I regret to inform you that we decided that in light of the referee reports, we cannot publish your manuscript in Communications Psychology.

We sent your manuscript back to the original reviewers (Referees #2 and #3 from the previous review cycle at Nature Communications). We also obtained signed comments from the authors of the work on which your study builds. Signed comments are not reviews. When a research contribution directly builds upon or criticizes an existing study, we solicit signed comments from the authors of the original research paper to ascertain whether their work is presented accurately. Potential remarks in the signed comments about whether the new contribution is novel and important, or whether it should be published or not, are not taken into account.

You will see that in the present case, the authors of the signed comments as well as Reviewer #3 disagree with the presentation of the target study. Because your study is conceptualized as a direct response to and correction of the previous paper, the criticism about the faithfulness of its presentation affects the evaluation of the degree of advance the new findings represent as well as the interpretability of the findings. This is why, in light of these comments, we cannot proceed further towards publication.

I am sorry that we cannot be more positive on this occasion and thank you for the opportunity to consider your work.

Best regards,

Troy Lui

Troy Lui, PhD
Associate Editor
Communications Psychology

REVIEWER COMMENTS:

Reviewer #2 (Remarks to the Author):

The authors have done an excellent job addressing my comments (and correcting my misunderstanding!).

I'm not sure DIC is really the gold standard model comparison metric (many people now use approximations of the marginal likelihood or cross-validation), but I think it's too much of a nitpick to ask the authors to redo the model recovery analysis.

Reviewer #3 (Remarks to the Author):

I reiterate that the present manuscript significantly mischaracterises the aims and findings of the Vassena et al. (2020) study it discusses. Firstly, it misrepresents the study's primary goal and hypotheses. The introduction frames the Vassena paper as if it were primarily proposing and testing a novel account focused on threshold adaptation driven directly by option values. However, the stated goal of Vassena et al. was clearly articulated in their abstract: "In this study, we derive predictions from three prominent accounts of dACC and test these with functional magnetic resonance imaging during value-based decision-making under time pressure". Their aim was thus to use fMRI to adjudicate between these existing models (CD, EVC, and PRO) within the context of a task requiring motivated control, not to primarily test a new value-based threshold theory.

Secondly, the paper misinterprets the role and significance of the DDM findings within the Vassena et al. study. It presents the observed lowering of decision thresholds for difficult choices as though it were the confirmation of a primary, a priori hypothesis central to the paper. In Vassena et al., however, the DDM analysis serves to characterise behaviour alongside the main fMRI investigation. The authors state they used DDM "to investigate the influence of value differences on key decision parameters". The threshold adjustment finding is reported as an observation ("We observed that ... the estimated decision threshold was lower than for those with clear value differences between options") and is subsequently interpreted post-hoc as reflecting control adjustments: "Adjustments of a decision threshold are a plausible target of proactive control processes and, in our experiment, can be explained as the need for subjects to decrease the decision threshold ... on difficult trials to ensure they respond before the response deadline". It was not framed as the test of the study's primary hypothesis.

Importantly, when the description of the work being critiqued diverges notably from the original publication—as evidenced by the direct quotes above and something that will be apparent to anyone familiar with the Vassena et al. paper—it undermines the critique that follows. I acknowledge that Vassena et al.'s tentative refutation of the EVC model hinges on demonstrating that their task engaged motivated control, for which they leverage the DDM findings as the key evidence. Whether or not these behavioural findings establish motivated control is thus a central point of discussion. However, the validity of Vassena et al.'s interpretation of their DDM results is orthogonal to the fundamental requirement that the present manuscript accurately describes their study's overall aims, methodology, and the specific comparisons performed. To critique Vassena et al.'s modelling approach or their interpretation of the DDM data with credibility, it remains essential first to represent their study's actual structure and claims accurately, otherwise, a reader may pause and question whether other aspects of the literature might also be misrepresented.

In this revision, the authors of the present manuscript have added the following passage to the introduction: "Having behaviorally validated this motivated control account, the authors go on to present it as an alternative to their preexisting account of neural data collected during the choice task (the PRO model;15). They show that the motivated control account, despite seeming to offer a good account of behavior, is unable to explain neural activity during this task (in contrast to the PRO model)."

This passage appears intended to describe how Vassena and colleagues compared different theoretical perspectives on dACC function. One possible interpretation is that "motivated control account" was used as shorthand for models emphasising value, choice difficulty, or control costs (like the CD and EVC models discussed in the paper), which Vassena et al. did indeed contrast with the surprise-based PRO model using their neural data. However, even with this reading of the potential underlying comparison, the specific phrasing chosen in the introduction unfortunately leads to an inaccurate interpretation of the Vassena et al. (2020) study's procedure and findings. By explicitly linking the account or model tested against the neural data to one supposedly "derived from DDM results" and first "behaviorally validated", the passage misrepresents the actual methodology. Vassena and colleagues did not formulate a new model based on their DDM findings to test against fMRI; rather, they directly compared the fMRI predictions of the pre-existing CD, EVC, and PRO models. It was primarily the CD model, not an account uniquely derived from the behavioural DDM results, that was shown to be inconsistent with the dACC activity patterns, as this passage implies.

Finally, I accept the authors' important clarification that collapsing thresholds do not equate to motivated control. This resolves my prior misunderstanding where I believed the manuscript's analyses supported Vassena et al.'s conclusions. By separating these concepts, I now see how the manuscript's analyses effectively question the specific claim of motivated control in the original paper that underlie their refutation of the EVC model.

Note on appeals: In exceptional circumstances, it is in authors' interest to appeal an editorial decision. More information on appeals is available here: <https://www.nature.com/commspsychol/submit/editorial-process#appeals>

Version 1:

Decision Letter:

Dear Harrison & Romy,

I have taken on handling of your manuscript during Troby's recent absence. On her behalf, I am delighted to say that we are happy, in principle, to publish a suitably revised version in Communications Psychology.

We therefore invite you to revise your paper one last time to address the remaining concerns of our reviewers and a list of editorial requests. At the same time we ask that you edit your manuscript to comply with our format requirements and to maximise the accessibility and therefore the impact of your work.

EDITORIAL REQUESTS:

SUBMISSION INFORMATION:

In order to accept your paper, we require the files listed here <https://www.nature.com/documents/commsj-file-checklist.pdf> .

OPEN ACCESS:

*** TRANSPARENT PEER REVIEW:** Communications Psychology uses a transparent peer review system. On author request, confidential information and data can be removed from the published reviewer reports and rebuttal letters prior to publication. If you are concerned about the release of confidential data, please let us know specifically what information you would like to have removed. Please note that we cannot incorporate redactions for any other reasons.

*** CODE AVAILABILITY:** All Communications Psychology manuscripts must include a section titled "Code Availability" at the end of the methods section. We require that the custom analysis code supporting your conclusions is made available in a publicly accessible repository at this stage; please choose a repository that generates a digital object identifier (DOI) for the code; the link to the repository and the DOI must be included in the Code Availability statement. Publication as Supplementary Information will not suffice.

*** DATA AVAILABILITY:**

Link Redacted

Best wishes,

Marika

Marika Schiffer, PhD
Chief Editor
Communications Psychology

REVIEWERS' COMMENTS:

Reviewer #3 (Remarks to the Author):

The authors have revised the manuscript to address my concern that it mischaracterised the aims and findings of Vassena et al. (2020). The current version resolves these issues and clarifies how Vassena et al. adapted the EVC model to their task. With these revisions, I believe that readers will be better able to connect their understanding of Vassena et al. with the explanation provided in the present manuscript. I therefore have no further comments.

Title: Commentary on Phantom controllers: Misspecified models create the false appearance of adaptive control during value-based choice

25 April, 2025

Authors: William H. Alexander^{1,2,3}

Eliana Vassena⁴

1. Center for Complex Systems & Brain Sciences, Florida Atlantic University
2. Department of Psychology, Florida Atlantic University
3. The Brain Institute, Florida Atlantic University
4. Donders Institute for Brain Cognition and Behaviour, Radboud University, Nijmegen, the Netherlands.

Corresponding author: William Alexander, Center for Complex Systems & Brain Sciences, Florida Atlantic University, 777 Glades Rd. Boca Raton, FL 33431 walexander@fau.edu

In their manuscript¹, Ritz, Froemer & Shenhav present an extensive reanalysis of behavioral data originally published in Vassena, Deraeve & Alexander(2020)² (hereafter VDA). While we are gratified by the attention the authors have devoted to our results, key aspects of our work are incorrectly characterized in their manuscript.

One such mischaracterization concerns the applicability of cognitive control to our task. Briefly, we developed a straightforward value-based choice task under time pressure, a manipulation that is specifically noted by the authors of one of the candidate models (Expected Value of Control; EVC) as eliciting markers of cognitive control³. There is a strong *normative* rationale for the involvement of cognitive control in time pressure manipulations: in order to meet a rapid response deadline, control-sensitive variables such as response threshold may be adjusted to encourage either increased or decreased response times^{4,5}. Within a trial, stimulus information (or metacognitive information derived from trial stimuli) can contribute to adaptive online control⁶ as Ritz et al. specifically acknowledge.

Ritz et al. contend that our use of value difference as the basis for updating control during a trial violates normative foundations of control. Specifically, they mistakenly suppose that value information in our simulated predictions of the EVC model is used to support two decisions, one decision about whether to adjust response thresholds, and a second decision regarding the identity of the more valuable option. According to their reading, having sufficient information about value differences between options to adjust a response threshold obviates the need to do so as subjects would already have the information needed to make a decision in the first place.

This reading of our simulations ignores two critical factors that place our task clearly within the bounds of normative control accounts. First, subjects are not simply ‘making a decision’ about which option to select. Due to the short response deadline, it is likely that subjects will have only incompletely integrated information regarding their options by the time they are required to generate a response^{7,8}. Given the incomplete information available to subjects, plausible control strategies within the trial may involve adjusting the response threshold⁹ to allow more time to integrate information, or invigorating a response for a trial that does not merit additional time for information integration to ensure a timely response.

This leads us to the second issue with Ritz et al.’s reading, namely that, in their view, stimulus values *per se* are not plausible metacognitive representations suitable for online control adjustments. This reading is misguided. First, between-option value differences are not synonymous with raw stimulus values – in our task, trials with equivalent value differences *between* options may nonetheless have very different values *within* each option. Second, (absolute) value differences between options are highly correlated with metacognitive representations (e.g., uncertainty or conflict) specifically endorsed by Ritz et al. as being appropriate for online control adjustments. Uncertainty reflects the variance amongst possible outcomes, which in our task is highest for large between-option value differences and lowest for small between-option value differences. Behavioral conflict¹⁰ reflects the coactivation of incompatible responses, which in our

task is highest for small between-option value differences and lowest for large between-option value differences. Our use of between-option value differences for generating predictions for the EVC model is supported insofar as they are not direct copies of individual values, as Ritz et al. appear to assume, but represent a summary description of the options on offer within a trial, i.e., a metacognitive representation. Furthermore, even if one does not accept this reasoning, given the correlation between value difference with variables that Ritz et al. specifically endorse as being acceptable for online adjustment of control, these other, “appropriate” variables can be swapped into our simulations of the EVC model without any change in the predictions we derive from them.

A second misrepresentation of VDA in Ritz et al. concerns the motivations underlying the study to begin with. As is clear from VDA, while neurocognitive models of control are frequently developed and tested under an assumption of *inhibitory* control, it was uncertain at the time whether these models’ predictions about brain activity could generalize to tasks in which control may be necessary for driving the execution of a response, i.e., motivated control.^{11,12} Using the speeded decision-making task, we derived predictions for brain activity using the published versions of a set of control models^{3,13,14}. To ensure our implementation of each candidate model fairly captured the predictions from each model, we additionally simulated patterns of brain activity the authors of each model had ascribed to their model. In particular, our implementation of EVC¹⁵ accurately reproduces activity recorded from anterior cingulate cortex that the authors of the EVC model themselves attribute to that model¹⁴.

Thus, from the outset, our explicit intent was to test established control models in a novel context, a key step in validating models¹⁶. Ritz et al., however, invent a straw model that they attribute to VDA and proceed to vigorously attack the predictions of that model. According to this straw model, VDA predicted *a priori* that response thresholds should be lower for difficult trials (i.e., those trials in which the expected value of each option is similar) than for easy trials. *In fact, no such prediction was made in VDA.*

To undermine the straw model, the authors focus on the results of a drift diffusion model¹⁷ (DDM) analysis reported in VDA in which we observe a lower response threshold for difficult trials relative to easy trials. While this result was *neither predicted nor required* by any of our model simulations, we offered an interpretation of this as being the product of the invigoration of a response to meet a looming response deadline. Ritz et al. attempt to undermine the validity of our actual predictions for the EVC model with a series of analyses showing that an elaborated DDM model incorporating decision thresholds and decision threshold collapse that vary as a function of trial difficulty better accounts for our behavioral data. Importantly for Ritz et al., the response threshold in these analyses is *higher* for difficult trials than easy trials. Based on this result, the authors reject the straw model they attribute to VDA – *a model we never endorsed* – and, by extension, the validity of our test of the predictions of the EVC model.

Critically, our EVC predictions did not rely on a specific control strategy, but identified those conditions in our task in which control should be licensed according to that model. As the

authors are aware¹⁸, control does not depend solely on the level of a response threshold, but also on *changes* to a response threshold: both lowering and raising a response threshold require that control be exerted. The authors' findings that both the initial response threshold and the rate at which that threshold collapses varies based on trial difficulty support the within-trial involvement of cognitive control, and, by extension, predictions we derive from the EVC model. The DDM analyses presented both in VDA as well as in Ritz et al. provide convergent behavioral evidence that control is indeed deployed during a trial in our task. Thus, the actual predictions we derived from the EVC model remain valid even if we stipulate that the DDM re-analyses presented in Ritz et al. are 100% correct.

Finally, the authors present a series of analyses in which data generated from a Leaky Competing Accumulator¹⁹ (LCA) model are analysed using DDM approaches, with the upshot being that, according to Ritz et al., our behavioral data could be derived from a generative process that does not incorporate control at all. The authors describe their LCA simulations as being “coarsely tuned” to the accuracy and reaction time of the participants in our study. We find the authors' implication that a control-free process could underly our data without undertaking formal model comparisons showing that the LCA in fact provides a better fit to be highly problematic. If the authors' contention is that our data were generated by something like a LCA, it should be straightforward enough to demonstrate that such a model better accounts for our behavioral data via quantitative fits rather than hand-tuned parameters that seem “close enough.”

Moreover, the authors are highly selective in the behavioral effects they choose to model using LCA. Besides our DDM analyses, VDA contained analyses of behavioral data demonstrating additional control effects during our task. Specifically, we showed sequential adjustment²⁰ and post-feedback slowing²¹, effects that have consistently been linked to control mechanisms. In short, the LCA analyses presented by Ritz et al. do not address the totality of the behavioral data presented in VDA, which convergently presents a clear picture of behavior in our task as being controlled both within and across trials.

In summary, in order to reject the conclusions of VDA with respect to the EVC model, Ritz et al. require that we ignore convergent lines of evidence and reasoning indicating that control is required for, and is in fact deployed in, our speeded decision-making task: 1) time pressure is generally known as a control-demanding manipulation, 2) there are strong normative reasons to suppose that our task elicits control within and across trials, and 3) Ritz et al. specifically note that online adjustment of control is supported by metacognitive representations equivalent to our use of between-option value differences. Furthermore, 4) both our own and the authors' DDM analyses identify difficulty-dependent adjustments in response threshold that 5) can only have occurred during the trial itself, and 6) our behavioral data exhibit additional, widely accepted effects associated with cognitive control. Finally, 7) we observed activity predicted by the PRO model of control in 8) dorsal anterior cingulate, a region of the brain strongly associated with cognitive control that 9) is also engaged by other control-relevant events (feedback, reward level) during our task. While none of these points is sufficient on its own, together they provide a

consistent and compelling account of how cognitive control is involved in our task. In contrast, the key piece of evidence Ritz et al. rely on to reject the presence of control in our task, and to preserve the EVC model, is a hand-tuned model that simulates a subset of our behavioral data.

References

1. Ritz, H., Frömer, R. & Shenhav, A. Phantom controllers: Misspecified models create the false appearance of adaptive control during value-based choice. *TBA*.
2. Vassena, E., Deraeve, J. & Alexander, W. H. Surprise, value and control in anterior cingulate cortex during speeded decision-making. *Nat. Hum. Behav.* 1–11 (2020) doi:10.1038/s41562-019-0801-5.
3. Shenhav, A., Botvinick, M. M. & Cohen, J. D. The Expected Value of Control: An Integrative Theory of Anterior Cingulate Cortex Function. *Neuron* **79**, 217–240 (2013).
4. Wu, C. M., Schulz, E., Pleskac, T. J. & Speekenbrink, M. Time pressure changes how people explore and respond to uncertainty. *Sci. Rep.* **12**, 4122 (2022).
5. Endres, D. N. *et al.* Stop the clock because I can't stop: time pressure, but not monitoring pressure, impairs response inhibition performance. *J. Cogn. Psychol.* **32**, 627–644 (2020).
6. Lieder, F., Shenhav, A., Musslick, S. & Griffiths, T. L. Rational metareasoning and the plasticity of cognitive control. *PLOS Comput. Biol.* **14**, e1006043 (2018).
7. Hellmann, S., Zehetleitner, M. & Rausch, M. Confidence Is Influenced by Evidence Accumulation Time in Dynamical Decision Models. *Comput. Brain Behav.* **7**, 287–313 (2024).
8. Teoh, Y. Y., Yao, Z., Cunningham, W. A. & Hutcherson, C. A. Attentional priorities drive effects of time pressure on altruistic choice. *Nat. Commun.* **11**, 3534 (2020).
9. Drugowitsch, J., DeAngelis, G. C., Angelaki, D. E. & Pouget, A. Tuning the speed-accuracy trade-off to maximize reward rate in multisensory decision-making. *eLife* **4**, e06678 (2015).
10. Botvinick, M., Braver, T. S., Barch, D. M., Carter, C. S. & Cohen, J. C. Conflict monitoring and cognitive control. *Psychol. Rev.* **108**, 624–652 (2001).
11. Vassena, E., Deraeve, J. & Alexander, W. H. Predicting Motivation: Computational Models of PFC Can Explain Neural Coding of Motivation and Effort-based Decision-making in Health and Disease. *J. Cogn. Neurosci.* **29**, 1633–1645 (2017).
12. Botvinick, M. & Braver, T. Motivation and Cognitive Control: From Behavior to Neural Mechanism. *Annu. Rev. Psychol.* **66**, 83–113 (2015).

13. Alexander, W. H. & Brown, J. W. Medial prefrontal cortex as an action-outcome predictor. *Nat Neurosci* **14**, 1338–1344 (2011).
14. Shenhav, A., Straccia, M. A., Cohen, J. D. & Botvinick, M. Anterior cingulate engagement in a foraging context reflects choice difficulty, not foraging value. *Nat. Neurosci.* **17**, 1249–1254 (2014).
15. Alexander, W. & Vassena, E. Selective application of theory does not replace formal model tests: a reply to Shenhav et al. Preprint at <https://doi.org/10.31234/osf.io/ugjrf> (2020).
16. Blohm, G., Kording, K. P. & Schrater, P. R. A How-to-Model Guide for Neuroscience. *eNeuro* **7**, (2020).
17. Ratcliff, R. & McKoon, G. The Diffusion Decision Model: Theory and Data for Two-Choice Decision Tasks. *Neural Comput.* **20**, 873–922 (2008).
18. Grahek, I., Leng, X., Musslick, S. & Shenhav, A. Control adjustment costs limit goal flexibility: Empirical evidence and a computational account. *bioRxiv* 2023.08.22.554296 (2025) doi:10.1101/2023.08.22.554296.
19. Usher, M. & McClelland, J. L. The time course of perceptual choice: the leaky, competing accumulator model. *Psychol. Rev.* **108**, 550–592 (2001).
20. Gratton, G., Coles, M. G. & Donchin, E. Optimizing the use of information: strategic control of activation of responses. *J Exp Psychol Gen* **121**, 480–506 (1992).
21. Notebaert, W. *et al.* Post-error slowing: An orienting account. *Cognition* **111**, 275–279 (2009).

Reviewer 1

I reiterate that the present manuscript significantly mischaracterises the aims and findings of the Vassena et al. (2020) study it discusses. Firstly, it misrepresents the study's primary goal and hypotheses. The introduction frames the Vassena paper as if it were primarily proposing and testing a novel account focused on threshold adaptation driven directly by option values. However, the stated goal of Vassena et al. was clearly articulated in their abstract: "In this study, we derive predictions from three prominent accounts of dACC and test these with functional magnetic resonance imaging during value-based decision-making under time pressure". Their aim was thus to use fMRI to adjudicate between these existing models (CD, EVC, and PRO) within the context of a task requiring motivated control, not to primarily test a new value-based threshold theory.

We appreciate this feedback, and have substantially revised our manuscript to address this. Our original intent was to focus our rebuttal on the core problematic claims from the original paper, but we agree that the original paper deserves the proper contextualization of its work.

As we now make clear, in pursuing its goal of comparing the neural predictions across different accounts of dACC function, the original paper constructed a novel account of EVC, which they attempted to behaviorally validate through their DDM analyses.

We want to be clear on this: Vassena et al. did not test an existing account of EVC. The authors say as much during the introduction: "Although successful in explaining dACC involvement in inhibitory control, it remains unclear whether these mechanisms generalize to motivated control." Therefore, in order to generate the model-based neural predictions that they attributed to EVC, they needed to verify the presence of the relevant cognitive processes. They are explicit about this in the results: "The predictions derived from the EVC model in our simulations are valid only if it can be demonstrated that our speeded decision-making task requires control." The model they generate also makes several assumptions that are not articulated in previous EVC accounts, such as having specific option values drive both decisions and control. After re-analysis, it is clear that their proposed model poorly accounts for the data, and that there are plausible alternative explanations. As they acknowledge in the quote above, this undermines their fMRI results.

We have now re-written most of the introduction based on the reviewer's recommendations.

Secondly, the paper misinterprets the role and significance of the DDM findings within the Vassena et al. study. It presents the observed lowering of decision thresholds for difficult choices as though it were the confirmation of a primary, a priori hypothesis central to the paper. In Vassena et al., however, the DDM analysis serves to characterise behaviour alongside the main fMRI investigation. The authors state they used DDM "to investigate the influence of value differences on key decision parameters". The threshold adjustment finding is reported as an observation ("We observed that ... the estimated decision threshold was lower than for those with clear value differences ... between options") and is subsequently interpreted post-hoc as reflecting control adjustments: "Adjustments of a decision threshold are a plausible target of proactive control processes and, in our experiment, can be

explained as the need for subjects to decrease the decision threshold ... on difficult trials to ensure they respond before the response deadline". It was not framed as the test of the study's primary hypothesis.

In their introduction, the authors do, in fact, motivate this particular control adjustment.

On control adjustments in value-based choice, they write:

Although value-based choice may not require explicit control requirements due to the increased incentive or salience of valuable options, it has been demonstrated that the addition of time pressure to such tasks produces effects consistent with cognitive control, including changes in the speed–accuracy trade-off and adjustments in decision parameters underlying choice behaviour. Specifically, increases in time pressure are correlated with decreases in the decision boundary of a diffusion process, a variable that is associated with the exertion of control and distinct from other factors, such as drift rate, that may influence response without recruiting control.

Later in the introduction, they write:

*As in typical cognitive control experiments, we anticipated that hard trials would require the deployment of additional control processes relative to easy trials. Specifically, **we expected that hard trials would require motivated control to generate a response within the response deadline.***

As we outlined above (re-iterating the authors' stated intentions), these behavioral tests are critical. They are the only verification supporting the imputed model-based predictions from the EVC model that form the basis of their fMRI analysis.

Importantly, when the description of the work being critiqued diverges notably from the original publication—as evidenced by the direct quotes above and something that will be apparent to anyone familiar with the Vassena et al. paper—it undermines the critique that follows. I acknowledge that Vassena et al.'s tentative refutation of the EVC model hinges on demonstrating that their task engaged motivated control, for which they leverage the DDM findings as the key evidence. Whether or not these behavioural findings establish motivated control is thus a central point of discussion. However, the validity of Vassena et al.'s interpretation of their DDM results is orthogonal to the fundamental requirement that the present manuscript accurately describes their study's overall aims, methodology, and the specific comparisons performed. To critique Vassena et al.'s modelling approach or their interpretation of the DDM data with credibility, it remains essential first to represent their study's actual structure and claims accurately, otherwise, a reader may pause and question whether other aspects of the literature might also be misrepresented.

We appreciate this concern, and agree that the previous manuscript did not sufficiently unpack the relevant context for the claims we made. We hope that the reviewer will find that our substantially revised introduction fills these gaps, and in doing so clarifies the implications and broader stakes of our findings.

In this revision, the authors of the present manuscript have added the following passage to the introduction: “Having behaviorally validated this motivated control account, the authors go on to present it as an alternative to their preexisting account of neural data collected during the choice task (the PRO model;15). They show that the motivated control account, despite seeming to offer a good account of behavior, is unable to explain neural activity during this task (in contrast to the PRO model).”

This passage appears intended to describe how Vassena and colleagues compared different theoretical perspectives on dACC function. One possible interpretation is that “motivated control account” was used as shorthand for models emphasising value, choice difficulty, or control costs (like the CD and EVC models discussed in the paper), which Vassena et al. did indeed contrast with the surprise-based PRO model using their neural data. However, even with this reading of the potential underlying comparison, the specific phrasing chosen in the introduction unfortunately leads to an inaccurate interpretation of the Vassena et al. (2020) study’s procedure and findings. By explicitly linking the account or model tested against the neural data to one supposedly “derived from DDM results” and first “behaviorally validated”, the passage misrepresents the actual methodology. Vassena and colleagues did not formulate a new model based on their DDM findings to test against fMRI; rather, they directly compared the fMRI predictions of the pre-existing CD, EVC, and PRO models. It was primarily the CD model, not an account uniquely derived from the behavioural DDM results, that was shown to be inconsistent with the dACC activity patterns, as this passage implies.

This highlights the importance of our revised introduction. The original authors did not test a pre-existing EVC account, making their behavioural verification critical for the interpretability of their imputed model-based predictions. This is clear from the authors’ stated intentions, highlighted above and is now reiterated, in the revised introduction.

There is important context here, that we have strenuously avoided wading into in this paper but will share with the reviewer for the sake of transparency. When Vassena et al. first shared their manuscript with the original EVC authors (Shenhav, Botvinick, Cohen) years ago, the EVC authors were very clear about their concerns about (1) the control mechanisms they attributed to EVC (i.e., response invigoration, which is not a part of existing EVC theories), and (2) the neural predictions from EVC theory (given that previous iterations of the model do not make explicit neural predictions). As detailed in a subsequent commentary (Shenhav et al., 2020), these assumptions were not endorsed by the authors of the EVC theory, and are in tension with qualitative predictions those authors have previously articulated (Shenhav et al., 2013; Shenhav et al., 2016).

Thus, while the introduction and abstract of the authors’ paper gives the impression that EVC’s predictions for this study can be asserted a priori, that is not actually the case. Instead, the EVC predictions in this paper very much depended on there being a justification for their control signals exactly the way that they assumed for their model-based predictions. Put differently, if Vassena et al. had not reported their DDM results, they would have had little basis for claiming that EVC makes a particular set of predictions for their task, and they would have been left to only refute the CD account (which had been previously refuted by work showing that dACC responds to factors other than value similarity during decision-making (Blair et al., 2006; Shenhav et al., 2018; Shenhav et al., 2014)). Indeed, it is

noteworthy that the authors only added the DDM analyses after being critiqued at previous journals for lacking a basis for their implementation of EVC.

We have, of course, omitted this historical context in our paper, and instead focus on the concerns most relevant to our analyses.

Finally, I accept the authors' important clarification that collapsing thresholds do not equate to motivated control. This resolves my prior misunderstanding where I believed the manuscript's analyses supported Vassena et al.'s conclusions. By separating these concepts, I now see how the manuscript's analyses effectively question the specific claim of motivated control in the original paper that underlie their refutation of the EVC model.

We're glad that our revisions have clarified this. We hope that our substantially revised introduction provides the necessary context for the importance of their mischaracterization of the cognitive and neural predictions of EVC theory.

Response to Original Authors

In their manuscript¹, Ritz, Froemer & Shenhav present an extensive reanalysis of behavioral data originally published in Vassena, Deraeve & Alexander(2020)² (hereafter VDA). While we are gratified by the attention the authors have devoted to our results, key aspects of our work are incorrectly characterized in their manuscript.

One such mischaracterization concerns the applicability of cognitive control to our task. Briefly, we developed a straightforward value-based choice task under time pressure, a manipulation that is specifically noted by the authors of one of the candidate models (Expected Value of Control; EVC) as eliciting markers of cognitive control³. There is a strong *normative* rationale for the involvement of cognitive control in time pressure manipulations: in order to meet a rapid response deadline, control-sensitive variables such as response threshold may be adjusted to encourage either increased or decreased response times^{4,5}. Within a trial, stimulus information (or metacognitive information derived from trial stimuli) can contribute to adaptive online control⁶ as Ritz et al. specifically acknowledge.

Ritz et al. contend that our use of value difference as the basis for updating control during a trial violates normative foundations of control. Specifically, they mistakenly suppose that value information in our simulated predictions of the EVC model is used to support two decisions, one decision about whether to adjust response thresholds, and a second decision regarding the identity of the more valuable option. According to their reading, having sufficient information about value differences between options to adjust a response threshold obviates the need to do so as subjects would already have the information needed to make a decision in the first place.

This reading of our simulations ignores two critical factors that place our task clearly within the bounds of normative control accounts. First, subjects are not simply 'making a decision' about which option to select. Due to the short response deadline, it is likely that subjects will have only incompletely integrated information regarding their options by the time they are required to generate a response^{7,8}. Given the incomplete information available to subjects, plausible control strategies within the trial may involve adjusting the response threshold⁹ to allow more time to integrate information, or invigorating a response for a trial that does not merit additional time for information integration to ensure a timely response.

This leads us to the second issue with Ritz et al.'s reading, namely that, in their view, stimulus values *per se* are not plausible metacognitive representations suitable for online control adjustments. This reading is misguided. First, between-option value differences are not synonymous with raw stimulus values – in our task, trials with equivalent value differences *between* options may nonetheless have very different values *within* each option. Second, (absolute) value differences between options are highly correlated with metacognitive representations (e.g., uncertainty or conflict) specifically endorsed by Ritz et al. as being appropriate for online control adjustments. Uncertainty reflects the variance amongst possible outcomes, which in our task is highest for large between-option value differences and lowest for small between-option value differences. Behavioral conflict¹⁰ reflects the coactivation of incompatible

responses, which in our task is highest for small between-option value differences and lowest for large between-option value differences. Our use of between-option value differences for generating predictions for the EVC model is supported insofar as they are not direct copies of individual values, as Ritz et al. appear to assume, but represent a summary description of the options on offer within a trial, i.e., a metacognitive representation. Furthermore, even if one does not accept this reasoning, given the correlation between value difference with variables that Ritz et al. specifically endorse as being acceptable for online adjustment of control, these other, “appropriate” variables can be swapped into our simulations of the EVC model without any change in the predictions we derive from them.

Thus far, Alexander and Vassena’s rebuttal focuses on several concerns with our characterization of the normativity of their model. Before we address these concerns, it’s important to note that our paper focuses primarily on the empirical support for their model rather than its normative basis. Even if we were to identify a normative basis for this model, the main thrust of our paper would still undercut the empirical support used to validate this model. Having said this, the rebuttal above fails to address the core issues that have been raised about VDA’s normative account from ourselves (Shenhav et al., 2020), as well as Reviewer 3’s response to our paper:

What I would consider the fundamental issue with using fixed boundaries is only mentioned by the current manuscript in the Discussion: the agent already seems to know the very values it is trying to identify. There is no way this can be literally correct. Thus, I think the authors of the current paper are overinterpreting what Vassena et al. are trying to show in their paper, and neither Vassena et al. nor any reader of that paper will literally interpret this analysis to mean that our brain can presciently use the results of an evidence accumulation process to adjust the process’s own parameters, right from the start.

[...]

It is quite puzzling that Vassena et al. didn’t explicitly use collapsing boundaries. Perhaps I myself am over-interpreting their work and this point truly needs to be explicitly made. However, as the literal interpretation is absurd, I doubt that it will be an “influential new theory,” as the authors of the current paper claim. Which studies have been influenced by this analysis?

It is true that their account could have replaced individual option values with a metacognitively accessible variable like uncertainty, but doing so would likely have also led to different predictions than those that VDA attributed to EVC. While it depends on the specifications of the model, past work focused on optimizing threshold around levels of conflict/uncertainty has reached the conclusion, normatively and empirically, that higher levels of conflict should lead to increases rather than decreases in threshold (Frank et al., 2015; for a review see Frömer & Shenhav, 2022). Indeed, Vassena recently co-authored a paper that came to this alternate conclusion about optimal control settings for VDA’s task (using a model other than EVC) (Vriens et al., 2025).

The key point here is that while VDA could have considered alternate formulations of EVC to account for control demands like conflict and even surprise (two factors that the EVC authors had explicitly argued should be considered in tasks like this), VDA instead

made a very narrow set of predictions and attributed those to EVC. If they had offered their threshold control account as one of many predictions that EVC could make in this setting, then their rebuttal above would be apt, but this would have also undercut their conclusion that EVC could not account for the observed neural data. For example, the brain plausibly monitors surprise to guide control adjustments — for example, the surprise-dependent slowing reported in their manuscript — so surprise signals in dACC are potentially consistent with EVC.

A second misrepresentation of VDA in Ritz et al. concerns the motivations underlying the study to begin with. As is clear from VDA, while neurocognitive models of control are frequently developed and tested under an assumption of *inhibitory* control, it was uncertain at the time whether these models' predictions about brain activity could generalize to tasks in which control may be necessary for driving the execution of a response, i.e., motivated control.^{11,12} Using the speeded decision-making task, we derived predictions for brain activity using the published versions of a set of control models^{3,13,14}.

This is incorrect by two accounts.

First, they misattribute the EVC optimization equation

$$signal(*) = \max_i[EVC(signal_i, state)]$$

as reflecting a neural prediction (“Finally, the EVC model states that dACC activity is proportional to the intensity of the optimal control signal (eq. 2).” (Vassena et al., 2020). This inconsistent with Shenhav et al (2013), which clearly states that signal* is an algorithmic property of the EVC computation (“Once it has been specified, the optimal control signal (signal(*)) is implemented and maintained by mechanisms responsible for the regulative component of control, which guide information processing in the service of task performance.”). Critically, Shenhav et al (2013) makes no commitments to the relationship between dACC activity and the identity of the control signal. It is unclear how the identity of the control signal — e.g., whether to regulate drift rate or threshold — would even result in changes to the univariate BOLD response.

Second, their specific equations linking effort to responses ('EVC1' and 'EVC2'), their critical embodiment of EVC in this paper, is nowhere in the EVC paper they cite.

$$P(O_1|effort_1) = \frac{1}{1 + \exp\left(-\left(\text{effort}_1 + \frac{V_1 - V_2}{\gamma}\right) \times \beta\right)}$$

To ensure our implementation of each candidate model fairly captured the predictions from each model, we additionally simulated patterns of brain activity the authors of each model had ascribed to their model. In particular, our implementation of EVC¹⁵ accurately reproduces activity recorded from anterior cingulate cortex that the authors of the EVC model themselves attribute to that model¹⁴.

This is misleading. First, because it suggests that the EVC model has previously provided a quantitative read-out of BOLD activity in dACC that can be compared with

empirical data, which is not the case. It is therefore not possible to compare quantitative fMRI predictions generated by VDA's EVC model with prior quantitative fMRI EVC predictions because the latter do not exist.

Setting this concern aside, the reviewers are raising a fair point that their implementation of the EVC model is able to capture neural correlates of choice difficulty that were observed in our 2014 study of foraging decisions, and which we noted at the time could be accounted for by the EVC theory. However, they omit the fact that this 2014 study also reported surprise-like effects observed in dACC activity, which we also attributed to EVC. The combination of conflict and surprise signals we observed in dACC during this value-based choice task perfectly foreshadowed the inverse W-like findings reported in VDA's 2020 paper.

Thus, from the outset, our explicit intent was to test established control models in a novel context, a key step in validating models¹⁶. Ritz et al., however, invent a straw model that they attribute to VDA and proceed to vigorously attack the predictions of that model. According to this straw model, VDA predicted a priori that response thresholds should be lower for difficult trials (i.e., those trials in which the expected value of each option is similar) than for easy trials. In fact, no such prediction was made in VDA.

Here, we quote VDA directly, from their introduction:

*As in typical cognitive control experiments, we anticipated that hard trials would require the deployment of additional control processes relative to easy trials. Specifically, **we expected that hard trials would require motivated control to generate a response within the response deadline.***

and:

Although value-based choice may not require explicit control requirements due to the increased incentive or salience of valuable options, it has been demonstrated that the addition of time pressure to such tasks produces effects consistent with cognitive control, including changes in the speed–accuracy trade-off and adjustments in decision parameters underlying choice behaviour. Specifically, increases in time pressure are correlated with decreases in the decision boundary of a diffusion process, a variable that is associated with the exertion of control and distinct from other factors, such as drift rate, that may influence response without recruiting control.

To undermine the straw model, the authors focus on the results of a drift diffusion model¹⁷ (DDM) analysis reported in VDA in which we observe a lower response threshold for difficult trials relative to easy trials. While this result was neither predicted nor required by any of our model simulations, we offered an interpretation of this as being the product of the invigoration of a response to meet a looming response deadline. Ritz et al. attempt to undermine the validity of our actual predictions for the EVC model with a series of analyses showing that an elaborated DDM model incorporating decision thresholds and decision threshold collapse that vary as a function of trial difficulty better accounts for our behavioral data. Importantly for Ritz et al., the response threshold in these analyses is higher for difficult trials than easy trials. Based on this result, the authors reject the straw model they attribute

to VDA – a model we never endorsed – and, by extension, the validity of our test of the predictions of the EVC model.

Critically, our EVC predictions did not rely on a specific control strategy, but identified those conditions in our task in which control should be licensed according to that model. As the authors are aware¹⁸, control does not depend solely on the level of a response threshold, but also on changes to a response threshold: both lowering and raising a response threshold require that control be exerted. The authors' findings that both the initial response threshold and the rate at which that threshold collapses varies based on trial difficulty support the within-trial involvement of cognitive control, and, by extension, predictions we derive from the EVC model. The DDM analyses presented both in VDA as well as in Ritz et al. provide convergent behavioral evidence that control is indeed deployed during a trial in our task. Thus, the actual predictions we derived from the EVC model remain valid even if we stipulate that the DDM re-analyses presented in Ritz et al. are 100% correct.

The reviewers are claiming here that the form of control and specific control demands (and by extension task incentives) are incidental to the predictions they generate from the EVC model. However, this is inconsistent with their use of DDM modeling to justify the specific dACC simulations, which then formed the basis of their model-based dACC analysis. Indeed, we've found that the predictions from EVC can strongly depend on the underlying assumptions that one makes about factors like controllability or valuation (Frömer et al., 2021; Leng et al., 2021; Musslick et al., 2018, 2019; Ritz et al., 2022).

This claim is also at odds with the reviewers' previous rebuttal paper arguing exactly the opposite (that the details of model implementation are critical to justifying one's predictions; (Alexander & Vassena, 2020)), and in their comments below on using the LCA as a mechanistic account of control.

Finally, we didn't find that the threshold "significantly" increased on difficult trials (credible intervals overlapped with zero), unlike the robust effects of overall value. The most striking trend for us was the inconsistency with the originally stated predictions. More importantly, we find that an LCA model — without control — can reproduce these putative threshold adjustments.

Finally, the authors present a series of analyses in which data generated from a Leaky Competing Accumulator¹⁹ (LCA) model are analysed using DDM approaches, with the upshot being that, according to Ritz et al., our behavioral data could be derived from a generative process that does not incorporate control at all. The authors describe their LCA simulations as being "coarsely tuned" to the accuracy and reaction time of the participants in our study. We find the authors' implication that a control-free process could underly our data without undertaking formal model comparisons showing that the LCA in fact provides a better fit to be highly problematic. If the authors' contention is that our data were generated by something like a LCA, it should be straightforward enough to demonstrate that such a model better accounts for our behavioral data via quantitative fits rather than hand-tuned parameters that seem "close enough."

The reviewers are conflating the statistical and scientific goals of the DDM and LCA models. The LCA is a well-established model of choice that, in previous work (Frömer et al., 2019), has been shown to capture value-based decision making and deadline choice. As in previous work (Frömer et al., 2019; Murphy et al., 2016), we used DDMs

for statistical analyses to avoid issues with parameter degeneracy that can arise in LCA fitting.

However, we have now revised the paper to include an LCA model that we fit to participants' behavior. We find that (1) this LCA model provides a better fit than DDM models, and (2) this fitted model continues to artifactually generate the signatures of control that VDA used to justify their EVC model.

These analyses make clear that the threshold effects reported by the original authors were weak justification for their neuroimaging analyses, given the plausible alternatives from standard decision frameworks.

Moreover, the authors are highly selective in the behavioral effects they choose to model using LCA. Besides our DDM analyses, VDA contained analyses of behavioral data demonstrating additional control effects during our task. Specifically, we showed sequential adjustment²⁰ and post-feedback slowing²¹, effects that have consistently been linked to control mechanisms. In short, the LCA analyses presented by Ritz et al. do not address the totality of the behavioral data presented in VDA, which convergently presents a clear picture of behavior in our task as being controlled both within and across trials.

The reviewers appear to be conflating the question of whether VDA's model assumptions were justified (the focus of our paper) with whether VDA's task demanded control. Our LCA analyses demonstrate that the behavior used to justify VDA's fMRI analysis can be produced by control-free decision processes.

Of course, as we and others have documented extensively (Frömer & Shenhav, 2022; Shenhav et al., 2013; Ullsperger et al., 2014), there are many other ways in which control might be used to modulate decision processes, including in response to error signals, conflict, and surprise. However, exploring these additional effects in this context is moot because none of these auxiliary assumptions were incorporated into VDA's EVC model. For example, VDA could have proposed an EVC model where post-feedback slowing was generated by the surprise signals they observed in the dACC.

In summary, in order to reject the conclusions of VDA with respect to the EVC model, Ritz et al. require that we ignore convergent lines of evidence and reasoning indicating that control is required for, and is in fact deployed in, our speeded decision-making task: 1) time pressure is generally known as a control-demanding manipulation, 2) there are strong normative reasons to suppose that our task elicits control within and across trials, and 3) Ritz et al. specifically note that online adjustment of control is supported by metacognitive representations equivalent to our use of between-option value differences.

While intuitively appealing, the logical case here comes apart on more careful examination.

The EVC theory makes predictions about how people allocate specific types of control under specific incentives and demands (Frömer et al., 2021; Leng et al., 2021; Shenhav et al., 2013; Shenhav et al., 2016). Indeed, in their implementation of EVC for their task, VDA had to specify a certain form of control (response invigoration) and certain values that deemed this control worthwhile (option values on the current trial). This was the entire basis of the

EVC-simulated dACC predictions for their task. The task could demand control in many other ways and for many other reasons (including across trials as they note), but their simulated dACC would be unaware of it, and instead would only vary its activity as a function of their specific form of motivated control.

Therefore, while VDA are certainly correct that decision-making is control-demanding for a variety of reasons (as we discuss elsewhere; (Frömer & Shenhav, 2022)), this misses the point. We are not arguing that their task doesn't demand any control. We are arguing that their task doesn't demand the specific kind of motivated control that would, on its own, justify the predictions that are central to their EVC refutation.

Conversely, if they had developed a model based on plausible assumptions about the influence of choice conflict and surprise on certain forms of control, we would not be in the position to refute these predictions based on their choice data. However, the predictions of such an EVC account would inconveniently mirror the predictions of their current PRO account, as we have previously noted (Shenhav et al., 2020).

Furthermore, 4) both our own and the authors' DDM analyses identify difficulty-dependent adjustments in response threshold that 5) can only have occurred during the trial itself, and 6) our behavioral data exhibit additional, widely accepted effects associated with cognitive control.

These claims raise the same concerns as above (regarding the specificity of model assumptions), but what the authors are also omitting here is that where these adjustments have been found in the past they have been associated with the opposite direction of effects as they claim are predicted by their EVC model (Frank et al., 2015).

Finally, 7) we observed activity predicted by the PRO model of control in 8) dorsal anterior cingulate, a region of the brain strongly associated with cognitive control that 9) is also engaged by other control-relevant events (feedback, reward level) during our task.

We have no objections to the evidence being provided for the PRO model. Evidence for this model doesn't necessitate evidence against other models that cannot be tested within a given experiment. Since our very first communications with VDA, the authors of the EVC model have not raised a single concern about the claims of support for the PRO model. The objection has only ever been to the predictions being wrongly attributed to us.

While none of these points is sufficient on its own, together they provide a consistent and compelling account of how cognitive control is involved in our task. In contrast, the key piece of evidence Ritz et al. rely on to reject the presence of control in our task, and to preserve the EVC model, is a hand-tuned model that simulates a subset of our behavioral data.

Reference list

- Alexander, W., & Vassena, E. (2020). Selective application of theory does not replace formal model tests: a reply to Shenhav et al. <https://doi.org/https://doi.org/10.31234/osf.io/ugjrf>
- Blair, K., Marsh, A. A., Morton, J., Vythilingam, M., Jones, M., Mondillo, K.,...Blair, J. R. (2006). Choosing the lesser of two evils, the better of two goods: specifying the roles of ventromedial prefrontal cortex and dorsal anterior cingulate in object choice. *J Neurosci*, 26(44), 11379-11386. <https://doi.org/10.1523/JNEUROSCI.1640-06.2006>
- Frank, M. J., Gagne, C., Nyhus, E., Masters, S., Wiecki, T. V., Cavanagh, J. F., & Badre, D. (2015). fMRI and EEG predictors of dynamic decision parameters during human reinforcement learning [10.1523/JNEUROSCI.2036-14.2015]. *J Neurosci*, 35(2), 485-494. <https://doi.org/10.1523/JNEUROSCI.2036-14.2015>
- Frömer, R., Dean Wolf, C. K., & Shenhav, A. (2019). Goal congruency dominates reward value in accounting for behavioral and neural correlates of value-based decision-making. *Nature Communications*, 10(1), 4926. <https://doi.org/10.1038/s41467-019-12931-x>
- Frömer, R., Lin, H., Dean Wolf, C. K., Inzlicht, M., & Shenhav, A. (2021). Expectations of reward and efficacy guide cognitive control allocation. *Nature Communications*, 12(1), 1030. <https://doi.org/10.1038/s41467-021-21315-z>
- Frömer, R., & Shenhav, A. (2022). Filling the gaps: Cognitive control as a critical lens for understanding mechanisms of value-based decision-making. *Neuroscience & Biobehavioral Reviews*, 134, 104483. <https://doi.org/https://doi.org/10.1016/j.neubiorev.2021.12.006>
- Leng, X., Yee, D., Ritz, H., & Shenhav, A. (2021). Dissociable influences of reward and punishment on adaptive cognitive control. *PLoS Comput Biol*, 17(12), e1009737. <https://doi.org/10.1371/journal.pcbi.1009737>
- Murphy, P. R., Boonstra, E., & Nieuwenhuis, S. (2016). Global gain modulation generates time-dependent urgency during perceptual choice in humans. *Nature Communications*, 7(1), 13526. <https://doi.org/10.1038/ncomms13526>
- Musslick, S., Cohen, J. D., & Shenhav, A. (2018). Estimating the costs of cognitive control from task performance: theoretical validation and potential pitfalls. Proceedings of the Annual Meeting of the Cognitive Science Society,
- Musslick, S., Cohen, J. D., & Shenhav, A. (2019). Decomposing individual differences in cognitive control: A model-based approach. Proceedings of the Annual Meeting of the Cognitive Science Society,
- Ritz, H., Leng, X., & Shenhav, A. (2022). Cognitive Control as a Multivariate Optimization Problem. *J Cogn Neurosci*, 34(4), 569-591. https://doi.org/10.1162/jocn_a_01822
- Shenhav, A., Botvinick, M. M., & Cohen, J. D. (2013). The Expected Value of Control: An Integrative Theory of Anterior Cingulate Cortex Function. *Neuron*, 79(2), 217-240. <https://doi.org/http://dx.doi.org/10.1016/j.neuron.2013.07.007>
- Shenhav, A., Cohen, J. D., & Botvinick, M. M. (2016). Dorsal anterior cingulate cortex and the value of control [Perspective]. *Nat Neurosci*, 19(10), 1286-1291. <https://doi.org/10.1038/nn.4384>
- Shenhav, A., Dean Wolf, C. K., & Karmarkar, U. R. (2018). The evil of banality: When choosing between the mundane feels like choosing between the worst. *J Exp Psychol Gen*, 147(12), 1892-1904. <https://doi.org/10.1037/xge0000433>

- Shenhav, A., Musslick, S., Botvinick, M. M., & Cohen, J. D. (2020). Misdirected vigor: Differentiating the control of value from the value of control. *PsyArXiv*.
<https://doi.org/10.31234/osf.io/5bhwe>
- Shenhav, A., Straccia, M. A., Cohen, J. D., & Botvinick, M. M. (2014). Anterior cingulate engagement in a foraging context reflects choice difficulty, not foraging value [Article]. *Nat Neurosci*, 17(9), 1249-1254. <https://doi.org/10.1038/nn.3771>
<http://www.nature.com/neuro/journal/v17/n9/abs/nn.3771.html#supplementary-information>
- Ullsperger, M., Fischer, A. G., Nigbur, R., & Endrass, T. (2014). Neural mechanisms and temporal dynamics of performance monitoring. *Trends Cogn Sci*, 18(5), 259-267.
<https://doi.org/10.1016/j.tics.2014.02.009>
- Vassena, E., Deraeve, J., & Alexander, W. H. (2020). Surprise, value and control in anterior cingulate cortex during speeded decision-making. *Nature Human Behaviour*.
<https://doi.org/10.1038/s41562-019-0801-5>
- Vriens, T., Vassena, E., Pezzulo, G., Baldassarre, G., & Silvetti, M. (2025). Meta-Reinforcement Learning reconciles surprise, value, and control in the anterior cingulate cortex. *PLOS Computational Biology*, 21(4), e1013025.
<https://doi.org/10.1371/journal.pcbi.1013025>